# Application of PLGA in Tumor Immunotherapy

**DOI:** 10.3390/polym16091253

**Published:** 2024-04-30

**Authors:** Jiashuai Wu, Xiaopeng Wang, Yunduan Wang, Zhe Xun, Shuo Li

**Affiliations:** 1Innovation Institute, China Medical University, Shenyang 110122, China; www773671@163.com (J.W.); 2022318105@cmu.edu.cn (X.W.); 2School of Intelligent Medicine, China Medical University, Shenyang 110122, China; wangyunduan@cmu.edu.cn; 3Liaoning Key Laboratory of Obesity and Glucose/Lipid Associated Metabolic Diseases, Health Science Institute, China Medical University, Shenyang 110122, China; 4Department of Biochemistry & Molecular Biology, School of Life Sciences, China Medical University, Shenyang 110122, China

**Keywords:** PLGA, tumor, immunotherapy, modification, drug delivery

## Abstract

Biodegradable polymers have been extensively researched in the field of biomedicine. Polylactic-co-glycolic acid (PLGA), a biodegradable polymer material, has been widely used in drug delivery systems and has shown great potential in various medical fields, including vaccines, tissue engineering such as bone regeneration and wound healing, and 3D printing. Cancer, a group of diseases with high mortality rates worldwide, has recently garnered significant attention in the field of immune therapy research. In recent years, there has been growing interest in the delivery function of PLGA in tumor immunotherapy. In tumor immunotherapy, PLGA can serve as a carrier to load antigens on its surface, thereby enhancing the immune system’s ability to attack tumor cells. Additionally, PLGA can be used to formulate tumor vaccines and immunoadjuvants, thereby enhancing the efficacy of tumor immunotherapy. PLGA nanoparticles (NPs) can also enhance the effectiveness of tumor immunotherapy by regulating the activity and differentiation of immune cells, and by improving the expression and presentation of tumor antigens. Furthermore, due to the diverse physical properties and surface modifications of PLGA, it has a wider range of potential applications in tumor immunotherapy through the loading of various types of drugs or other innovative substances. We aim to highlight the recent advances and challenges of plga in the field of oncology therapy to stimulate further research and development of innovative PLGA-based approaches, and more effective and personalized cancer therapies.

## 1. Introduction

Cancer remains a formidable global health challenge, with its pervasive impact afflicting millions of lives worldwide [1]. Traditional modalities of cancer treatment, including chemotherapy and radiation therapy, are frequently constrained by issues such as systemic toxicity and the emergence of drug resistance [2,3]. In contrast, immunotherapy has emerged as a promising frontier in oncology, offering a potentially transformative approach to cancer treatment that may circumvent these limitations [4].

Biodegradable polymers have increasingly become a focal point within the realm of medical materials research. Their applications in the field of immunotherapy are particularly noteworthy, serving as vehicles for immunomodulatory drugs, constituents of anti-cancer vaccines, and as targeted delivery systems for antigens, thereby activating the immune response [5,6,7,8]. Among these polymers, poly(lactic-co-glycolic acid) (PLGA) stands out as an emerging material that has shown remarkable promise in performance, and the application of PLGA in the immunotherapy of cancer is highlighted in this article [9]. In addition, we have employed bibliometric methods, yielding a set of quantifiable outcomes that contribute to the depth of our research.

PLGA’s tunable properties, including degradation rate, mechanical strength, and drug release kinetics, make it an ideal candidate for addressing the challenges associated with cancer treatment [10,11,12]. Its unique immunogenicity, histocompatibility and Interaction with the immune system are its basic properties in immunotherapy.

At present, the fabrication of PLGA particles is facilitated by a variety of well-established techniques, such as: Single emulsion-solvent evaporation, double emulsion-solvent evaporation, solvent-precipitation, and spray drying. Each of these methodologies presents its own set of advantages and limitations, as well as a specific scope of application. Consequently, researchers are empowered to select the most fitting approach aligned with their unique research objectives. Moreover, the versatility of PLGA allows for targeted modifications that can alter its targeting capabilities, immunogenicity, and other fundamental properties. Such modifications are instrumental in the development of strategies for immune modulation and targeted delivery to tumors, thereby enhancing the therapeutic potential of PLGA-based formulations in cancer treatment.

PLGA plays an important role in cancer immunotherapy, and its main mechanisms include drug delivery and immune regulation. By encapsulating traditional anticancer drugs, such as paclitaxel and vinorelbine, etc., PLGA is able to enhance the anti-tumor effect of the drugs and deliver them precisely to the tumor site, reducing the damage to healthy tissues. Simultaneously, PLGA’s versatility extends to encapsulating cytokines, antigens and other immunomodulatory substances to regulate the activity of the immune system. For example, PLGA can encapsulate antigens to promote the activation of immune cells and enhance immune responses against tumor cells; In addition, PLGA can also encapsulate immunomodulators to inhibit the expression of immunosuppressive factors, thereby enhancing the ability of immune cells to attack tumors. Despite the potential of PLGA in cancer immunotherapy, PLGA faces certain challenges in cancer immunotherapy: limited drug loading capacity, difficult to regulate drug release rate in vivo, immunogenicity and immunostimulatory effects on the body, and lack of long-term stability in vivo. Nevertheless, ongoing advancements in synthesis techniques and the incorporation of surface modifications offer promising avenues to address these limitations and enhance the therapeutic potential of PLGA in cancer treatment.

The application of PLGA nanoparticles in cancer therapy addresses several limitations of conventional treatments such as chemotherapy, which often suffer from uneven drug distribution, severe side effects, and rapid drug elimination. PLGA nanoparticles provide a controlled and sustained release mechanism, reducing toxicity to normal tissues while increasing drug concentration at the tumor site, thus enhancing treatment efficacy [13]. PLGA nanoparticles are biocompatible and biodegradable, with degradation products that can be naturally metabolized, reducing the risk of long-term toxicity. Additionally, the ratio of lactic to glycolic acid in PLGA can be adjusted to tailor the drug release rate for personalized treatment [14]. As a carrier for immunotherapeutic agents, PLGA nanoparticles can simulate the effect of multiple injections through pulsatile release, enhancing immune responses, inhibiting tumor growth, and prolonging survival without the need for repeated injections, thereby improving patient compliance and reducing the risk of treatment-associated metastasis [15]. Surface modification with targeting ligands, such as antibodies or peptides, allows for specific targeting of tumor cells, enhancing treatment efficacy and reducing systemic side effects [16]. The biocompatibility and biodegradability of PLGA nanoparticles ensure their safety in the body, with studies showing complete degradation and clearance inflammation or other toxic reactions [9]. These advantages position PLGA nanoparticles as a promising approach in cancer therapy, offering patients more effective, safer, and more convenient treatment options.

In the field of biomedical applications, the terms PLGA Nanoparticles, Microparticles, and Nanopolymers are often used to delineate materials derived from PLGA. While these terms are often used to describe PLGA-based constructs, they denote different scales and applications. PLGA Nanoparticles are defined by their nanoscale size and are commonly utilized in drug delivery and imaging. Microparticles, larger in size, are typically used for drug release and as scaffolds in tissue engineering. Nanopolymers, a broader category, may encompass nanofibers and other nanostructures, offering a wide range of applications in the biomedical field.

Overall, this review provides a comprehensive overview of the various applications of PLGA in cancer treatment. We will discuss the immunological basis of PLGA in immunotherapy, the synthesis and preparation of PLGA, and the application of PLGA in cancer. By elucidating recent advancements alongside the challenges faced in this field, we aim to inspire further research and development of innovative PLGA-based approaches for effective and personalized cancer therapy.

## 2. Immunological Basis of PLGA in Immunotherapy

To fully harness the therapeutic potential of PLGA nanoparticles (NPs) in immunotherapy, a profound comprehension of their immunological foundation is imperative. This necessitates a detailed exploration of the biodistribution of NPs within the body, their cellular uptake mechanisms, and the intricacies of their interactions with immune cells. The optimization of NP design parameters could significantly augment their targeting efficacy and immune-stimulating properties, thereby enhancing the overall therapeutic impact of immunotherapy.

A thorough understanding of the interplay between NPs and the immune system is paramount for accurately assessing their metabolic fate and clearance profiles. This knowledge is crucial for predicting the long-term immunological consequences and potential adverse effects of PLGA NPs. Furthermore, insights into how NPs modulate immune cell functions and the regulation of immune responses are invaluable for the strategic refinement of immunotherapeutic protocols.

The objective of this part is to delineate the mechanistic pathways through which PLGA NPs exert their effects in immunotherapy. By doing so, we aim to establish a robust scientific framework that underpins their clinical translation and application, thereby facilitating the development of more effective and safer immunotherapeutic strategies.

The primary immunological mechanisms of PLGA involve inducing antigen-presenting cells (APCs), such as dendritic cells (DCs) and macrophages, to phagocytose antigens. This improves humoral and cellular immune responses by enhancing antigen presentation on MHC class I and II molecules [17,18]. The process involves three main steps: endosomal escape, proteasomal processing, and antigen loading in the endoplasmic reticulum (ER) and/or endosomes [19]. Upon recognition by the immune system, PLGA has been observed to modulate inflammatory responses, mitigate neuroinflammation following central nervous system (CNS) injury, and reprogram lipopolysaccharide-activated microglia from the pro-inflammatory phenotype to the anti-inflammatory phenotype [20]. Furthermore, PLGA has demonstrated the capacity to suppress the activity of immune cells within the CNS and augment their accumulation in the spleen [21]. PLGA affects immune responses through various mechanisms, such as enhancing antigen presentation, regulating inflammatory responses, promoting the immunological effects of antigens, and interacting with immune cells (Figure 1). Given these properties, PLGA emerges as a promising candidate for its application in the realms of immunotherapy, vaccine development, and the regulation of inflammatory conditions.

The interactions between PLGA nanoparticles and immune cells significantly influence both short-term and long-term immunological responses, and understanding these interactions is vital for enhancing the design and clinical application of PLGA-based immunotherapy strategies. In the short term, PLGA NPs can enhance antigen presentation, rapidly activating immune cells such as dendritic cells and T cells [22]. Additionally, immunoadjuvants like CpG oligonucleotides, when delivered via PLGA NPs, can quickly activate natural killer cells and T cells, initiating a swift immune response [23]. Over the long term, PLGA NPs can promote the generation of memory T cells, providing lasting immune protection against tumor reoccurrence [24]. Moreover, well-designed PLGA NPs can prevent immune tolerance to therapeutic antigens, ensuring the longevity of the immune response [25]. This knowledge can be utilized to strengthen PLGA-based immunotherapy strategies through targeted delivery, combination therapies, and sustained release systems [26,27,28]. In clinical applications, personalized treatments, postoperative therapies, and safety assessments are key directions for realizing the potential of PLGA NPs, enhancing the effectiveness of cancer treatment [29,30].

Understanding the immunological mechanisms of PLGA nanoparticles is crucial for optimizing their effectiveness in immunotherapy. The immunological mechanisms include the interaction of PLGA nanoparticles with antigen-presenting cells (APCs), their role in enhancing antigen presentation, and their ability to modulate immune responses. By elucidating these mechanisms, the design of PLGA nanoparticles can be tailored to improve their biocompatibility, targeting, and immune-stimulating properties. For example, surface modifications can enhance the interaction of nanoparticles with specific immune cells, thereby improving their therapeutic efficacy. Additionally, understanding the degradation products of PLGA and their impact on the immune system allows for the development of strategies to mitigate potential adverse immunological responses, ensuring the safety and effectiveness of PLGA nanoparticles in clinical applications. This knowledge can guide the design of multimodal nanoparticles capable of drug delivery, immune activation, and modulation of the tumor microenvironment, thereby enhancing the overall efficacy of immunotherapy.

PLGAs hold significant potential for applications in immunotherapy due to their unique properties, including charge, size, and morphology. These properties are known to exert a profound influence on the in vivo behavior of NPs, including their distribution, cellular uptake, and immunostimulatory capabilities. Consequently, these characteristics can modulate the immunological mechanisms and therapeutic efficacy of PLGA NPs. Given the critical role of these properties, it is imperative to undertake a thorough investigation into how the charge, size, and morphology of PLGA NPs impact immunotherapeutic outcomes. Such a study would not only enhance our understanding of the underlying principles but also inform the strategic design and optimization of PLGA NPs for use in immunotherapy. This knowledge is essential for advancing the development of more effective and targeted immunotherapies.

The surface charge of PLGA NPs markedly influences their immunological mechanisms, affecting their biodistribution, cellular internalization, and immunomodulatory functions. Nanoparticles bearing a positive charge are more inclined to be internalized by the negatively charged cell membranes, whereas those with a negative charge may interact more readily with positively charged cell surfaces. This electrostatic interaction can modulate the targeting efficacy and immunomodulatory potential of NPs within the context of immunotherapy. In the design of NPs for therapeutic applications, it is imperative to contemplate the influence of surface charge on the therapeutic outcome. Research has demonstrated that negatively charged PLGA NPs can interact with scavenger receptors present on circulating immune cells, thereby modulating immune cell function. This interaction may lead to the reprogramming of immune cells, influencing their trafficking to the spleen, or dispersing circulating immune cells, such as inflammatory monocytes and neutrophils, from the site of injury, indirectly mitigating immune pathology in inflamed regions [21]. Highly negatively charged PLGA NPs could also reduce immune cell infiltration, thereby lowering secondary damage resulting from inflammatory responses. PLGA may cause severe immune toxicity through hemolysis or platelet aggregation due to the electrostatic interactions of positively charged NPs with cell membranes, the interactions of hydrophobic NPs with the reticuloendothelial system, or the aggregation of serum proteins caused by small cationic NPs [31]. When formulating NPs for immunotherapy, it is crucial to balance the therapeutic benefits against the potential risks associated with NP surface charge. This balance aims to maximize therapeutic efficacy while minimizing adverse reactions, ensuring the safe and effective application of PLGA NPs in immunotherapeutic strategies.

The size of PLGA NPs is a determinant factor in their immunological mechanisms, particularly within the context of immunotherapy. Recent research underscores the pivotal role of particle size in eliciting antigen-specific T cell responses and modulating interactions with the immune system. A study led by Natalia Muñoz-Wolf examined PLGA particles with sizes varying from 50 nm to 30 μm, revealing that particle size is instrumental in the induction of antigen-specific CD8+ T and Th1 cell responses, without significantly affecting antibody production. The size range of 50–60 nm was identified as optimal for stimulating specific Th1 and CD8+ T cell responses to subunit protein antigens, and also for the induction of reactive oxygen species (ROS), which augments the cytotoxic potential against tumor cells. Vaccination with 50 nm particles was shown to elicit a sustained functional response, conferring protection against B16-ovalbumin (OVA) melanoma in mice [32]. Utilization of smaller particles may enhance uptake by immune cells and intracellular presentation of antigens, leading to a stronger immune response [31]. Furthermore, smaller particles may be more effective in penetrating tissue barriers, such as mucosal barriers, thereby improving interaction with the immune system. Nusaiba K Al-Nemrawi discovered that PLGA NPs of 300 nm size are more effective in promoting antigen-specific T cell responses compared to particles larger than 1 μm [31]. This finding underscores the importance of particle size in modulating immune responses. [5] Research conducted by Ryan M Pearson and colleagues on autoimmune diseases suggests that PLGA NPs with a size of 500 nm are the most suitable [33]. These NPs have a greater degree of internalization and affinity for binding to immune cells [21]. Additionally, Eiji Saito and colleagues found that PLGA-H (high molecular weight PLG) particles have a higher binding ability with immune cells. Compared to low molecular weight PLGA particles, PLGA-H particles have a higher affinity for neutrophils in the blood, which restricts immune cell activity in the central nervous system and increases accumulation in the spleen [34]. In summary, the size of PLGA NPs significantly influences their immunotherapeutic effectiveness. Smaller particles, particularly those in the 50–60 nm range, are more readily phagocytosed by immune cells, leading to more efficient antigen presentation and a robust immune response. Conversely, larger PLGA particles may engage immune cells through distinct binding mechanisms. When designing PLGA NPs for immunotherapy, it is imperative to consider the impact of particle size on therapeutic efficacy. A comprehensive understanding of this parameter is crucial for optimizing the performance and safety of PLGA NPs in clinical applications.

In addition, the morphology of PLGA NPs could also significantly influences their immunological mechanisms. In the realm of biomaterials and drug delivery, the morphology of biomimetic particles has emerged as a key parameter that can impact their interaction with biological systems [35]. Particle morphology is critical in reducing phagocytic endocytosis, cellular uptake, and altering the biodistribution of drug delivery carriers [36,37,38]. For instance, rod-shaped PLGA NPs exhibit a higher affinity for neutrophil absorption compared to their spherical counterparts, suggesting that the elongated structure may enhance therapeutic targeting of neutrophils [39]. Future use of this particle shape-based targeting of neutrophils will require researchers to obtain biocompatible and biodegradable non-spherical particles easily. Furthermore, it is worth noting that the shape of PLGA NPs may have varying effects on their uptake and internalization by immune cells, which in turn could impact their efficacy in immunotherapy. Kavya Sree Maravajjala and others found that compared to linear PLGA, star-shaped PLGA pH-responsive NPs demonstrated improved pH-dependent drug release and increased permeability in a complex breast cancer spheroid model (breast cancer cells and macrophages) [40]. Hence, it is imperative for forthcoming researchers to consider the influence of nanoparticle morphology on the therapeutic efficacy of immunotherapy, with the aim of designing more effective treatment strategies.

While PLGA nanoparticles (NPs) hold substantial promise in the field of immunotherapy, their clinical application is accompanied by a spectrum of challenges that necessitate further investigation and innovation. Comprehensive research is essential to elucidate the intricate immunological mechanisms of these NPs, which include their biodistribution, cellular uptake, immune regulation, and interactions with immune cells. A thorough understanding of these mechanisms is vital for the optimization of nanoparticle design, with the aim of enhancing their targeting capabilities and immunomodulatory effects. The biocompatibility and biodegradability of PLGA NPs must be carefully considered to ensure minimal adverse effects and to facilitate their safe integration into the human body. Additionally, overcoming technical hurdles in the preparation and production of NPs is paramount. This includes achieving consistency in particle size and shape, which are known to significantly influence the NPs’ immunological performance and therapeutic outcomes. Further research endeavors, coupled with advancements in nanotechnology, are imperative to surmount these challenges. The goal is to pave the way for the broad application of PLGA NPs in immunotherapy, thereby harnessing their full potential in the treatment of various diseases, including cancer.

We use bibliometric methods to analyze the immunologic mechanism of PLGA. We collected 6107 PLGA articles from the Web of Science database by PLGA (Topic) AND immun* (Topic), and finally analyzed a total of 2753 articles from 2018 to 2024 (Figure 2). The data indicate the importance of the immunomodulatory effects of PLGA in tumor treatment and demonstrate the research hotspots of immune cells such as DCs and Macrophages and s cytokines.

## 3. Surface Modification of PLGA

At present, PLGA could be one of the most widely used biodegradable polymers is characterized by its strong modification ability, researchers through nanotechnology modification, functionalization modification, bilayer structure modification, biodegradable polymer modification, nanopore modification, surface modification and other different modification methods on PLGA, so that the physical and chemical properties, immunogenicity, histocompatibility of PLGA could be changed, so that it could be applied to more fields. The distinct modification methods, each with its unique mechanism, are detailed as follows (Figure 3), showcasing the multifaceted approaches to optimizing PLGA for specific applications within the biomedical sector.

### 3.1. Surface Modification of Biomaterials

#### 3.1.1. Decreasing the Immunorecognition of PLGA and Increasing Immunocompatibility

Surface modification of PLGA microparticles with biocompatible materials can significantly diminish their immunorecognizability by the in vivo immune system. By integrating biocompatible molecules onto the surface of PLGA microparticles, the probability of immune system recognition can be reduced. This modification strategy is critical for enhancing the stealth characteristics of the particles, potentially improving their biocompatibility and reducing the risk of immune-mediated adverse reactions.

In the modification using biomaterials, antibodies could be used for modification, and by modifying antibody molecules on the surface of PLGA particles, the recognition and binding of specific antigens could be realized, thereby improving the targeting and specificity of the particles. This targeted surface modification facilitates more effective interactions between PLGA microparticles and tumor cells while minimizing immune system interference [41]. In addition, the use of cell membrane modification of PLGA is also a feasible method, and the membrane fragment derived from immunocompatible cell membrane, or the entire cell membrane could be modified on the surface of PLGA particles, which could increase the immunocompatibility of PLGA particles [42,43]. This surface modification could mimic the characteristics of its own cells, reducing the immune system’s recognition and attack of particles. Y. Wang et al. prepared macrophage membrane-coated biomimetic NPs (MM/RAPNPs) copolymers by modifying PLGA with macrophage membrane (MM), and determined their histocompatibility, and the results showed that the prepared MM/RAPNPs exhibited good biocompatibility in mice. Due to MM coating modifications, NPs potently inhibit macrophage phagocytosis and target activated endothelial cells in vitro. This modification helps to reduce the uptake of phagocytes (e.g., macrophages) and improve cycle time in the body [44]. This coating modification is instrumental in diminishing phagocytic uptake by immune cells, such as macrophages, and prolonging the circulation time of the nanoparticles within the body.

#### 3.1.2. Regulating the Immune Response

Surface modification through biomaterials could alleviate the immunosuppressive regulation of PLGA in the human body, such as unmodified PLGA particles may trigger inflammation and immune response of the immune system, and surface modification could alleviate this immune regulation. Through the immunological mechanism, we can predict that some modification of various biological factors can regulate the immune response. Antibody modification: By carrying antibody molecules on the surface of PLGA particles, it is possible to achieve a regulatory effect on immune cells. For example, modifying antibodies with immunosuppressive functions, such as anti-CD47 antibodies, could inhibit the phagocytosis of macrophages, thereby reducing immune and inflammatory responses. Immunosuppressive molecular modification: Modification of immunosuppressive molecules on the surface of PLGA microparticles could increase the immunosuppressive properties of microparticles. For example, modifying immunosuppressive factors such as TGF-β and IL-10 could inhibit the activation and inflammatory response of immune cells to achieve the effect of immunosuppression [45]. Cell membrane modification: modifying PLGA microparticles with membrane fragments or whole membranes derived from immunosuppressive cells can endow the particles with biomimetic properties that regulate immune responses. For instance, incorporating fragments of regulatory T cell membranes can induce immune tolerance and immunosuppressive outcomes. The functionalization of PLGA particle surfaces with immunosuppressive ligands allows for the binding to specific receptors on immune cells, thereby conferring immunosuppressive effects. For example, the modification with ligands such as PD-1 or CTLA-4 can bind to their respective receptors on immune cells, inhibiting T cell activation and subsequent immune responses [46].

However, it should be noted that the application of these modification strategies may modulate or attenuate the body’s immune response to a certain degree. If not meticulously controlled, such interventions could precipitate immune dysregulation or even precipitate diseases of the immune system. Consequently, a judicious approach is paramount in the clinical application of PLGA modification. It is imperative to take a holistic view that encompasses a comprehensive assessment of the patient’s immune status and the extent of PLGA modification [27]. This precautionary stance ensures that the immune response is neither over-suppressed nor under-stimulated, thereby maintaining a balanced and beneficial therapeutic effect.

Further modifications could increase its immunomodulatory function. By modifying APCs mimic molecules on the surface of PLGA particles, the function of APCs could be mimicked, activating and regulating the activity of immune cells. These modified molecules could include MHC molecules, costimulatory molecules, and inflammation regulators [47].

Immunostimulatory Molecule Modification: Immunostimulatory molecules are modified on the surface of PLGA microparticles to activate immune cells and regulate immune responses. For example, modifying immunostimulatory molecules such as TLR agonists, etc., could enhance antigen presentation, T cell activation, and immune response [48]. This modification strategy can be particularly effective in enhancing the immunogenicity of vaccine formulations and therapeutics that target the immune system.

#### 3.1.3. Increasing Antigen Adjuvant Capacity

Surface modifications of PLGA nanoparticles can significantly enhance their capacity to serve as antigenic adjuvants, a development with profound implications for vaccine design and immunotherapy. In a pivotal study, R. Han and colleagues encapsulated the model antigen ovalbumin (OVA) within protamine-modified PLGA NPs. These NPs were utilized to stimulate dendritic cells (DCs) derived from mouse bone marrow, demonstrating that protamine-coated PLGA NPs could augment the cross-presentation of encapsulated exogenous antigens. This effect was attributed to the promotion of antigen uptake and facilitation of lysosomal escape, which are critical for initiating an effective immune response [49]. 

PR Hartmeier et al. reported that PLGA NPs modified with biotin could deliver proteins and stimulate specialized APCs by adsorption [49]. 

Q. Liu and colleagues developed pH-responsive PLGA NPs using ammonium bicarbonate, which exhibited rapid intracellular antigen release behavior within APCs. These NPs acted as antigen release promoters in DCs when co-encapsulated with antigens such as OVA. Upon DC uptake, the pH-responsive PLGA NPs enabled antigens to escape from lysosomes into the cytoplasm, allowing for cross-presentation. Concurrently, these NPs induced the upregulation of costimulatory molecules and stimulated cytokine production. This multifaceted approach led to enhanced lymphocyte activation, increased generation of antigen-specific CD8+ T cells with stronger cytotoxic capacity, boosted antigen-specific antibody production, and improved the generation of memory T cells, thereby providing robust protection against reinfection. These findings suggest that appropriately modified PLGA NPs can elicit potent cellular immune responses and offer antibody-mediated protection [50].

### 3.2. Surface Modification of Chemical Material

Polyethylene glycol (PEG) modification is a widely adopted strategy for creating a PEG-rich surface layer on PLGA particles. This is achieved through the covalent attachment of PEG molecules to the particle surface, which confers several advantages for the application of these particles in biological systems. The PEG layer acts as a barrier to protein adsorption, mitigating the recognition and clearance by the immune system, and consequently reducing the immunogenicity associated with PLGA particles. This immunomodulatory effect of PEGylation can also enhance the stability and prolong the circulation time of PLGA particles in the body [51]. In addition, the selection of molecules with low immunogenicity as surface modifiers could reduce the immunorecognition of PLGA particles. For example, the selection of molecules such as polymers, carbohydrates, or lipids with low immunogenicity for modification could reduce the immune system’s response to PLGA microparticles [52,53]. This approach is particularly valuable in the design of nanoparticles intended for drug delivery and vaccine development, where minimizing immune recognition is critical for maintaining particle integrity and functionality.

P. Gu et al. developed three OVA loaded PLGA NPs with different surface charge and antigen loading modes, negatively charged antigens (Angelica sinensis polysaccharide (ASP)-PLGA/OVA), polyethyleneimine (PEI)-coated antigen (ASP-PLGA/OVA-PEI), and PEI-coated antigen (ASP-PLGA-PEI-OVA) NPs to study how the surface charge and antigen loading patterns of NPs affect immune responses. The results showed that both PEI-coated (positively charged) NPs facilitated antigen escape from endosomes compared to negatively charged NPs, resulting in cross-presentation of cytoplasmic antigen delivery [54,55].

Similarly, C. Song et al. found that PEI-coated PLGA (OVA) NP was efficiently internalized by phagocytosis in DCs or macrophage phagocytosis and induced efficient cross-presentation of antigens on MHC class I molecules through endosomal escape and lysosomal processing mechanisms [56].

Different experiments have found that PLGA modified by PEI could effectively promote the escape and cross-presentation of encapsulated drugs or antigens from endosomes. 

On the whole, different surface modification techniques of PLGA, including nanotechnology and biodegradable polymer modifications, significantly impact its physicochemical properties and suitability across various biomedical applications. Nanotechnology modifications, such as controlling the size and shape of PLGA at the nanoscale, can enhance hydrophilicity, reduce nonspecific protein adsorption and cell adhesion, thereby improving the circulation time and drug delivery efficiency of PLGA nanoparticles in the body [57]. Surface modifications with specific ligands or antibodies enable targeted delivery to particular cells or tissues, enhancing treatment specificity and reducing side effects [53]. For instance, aptamer-conjugated PLGA nanoparticles showcase the potential of ligand modification for targeted cancer therapy. Biodegradable polymer modifications, such as the covalent attachment of polyethylene glycol (PEG), can create a “stealth surface” on PLGA microspheres, reducing immune recognition and clearance, and prolonging their half-life in the body [58]. These modifications can also adjust the degradation rate of PLGA and the kinetics of drug release to suit different therapeutic needs. For example, altering the ratio of lactic to glycolic acid in PLGA copolymers can tailor the degradation rate and drug release profile [59]. These modifications positively affect drug delivery by enhancing bioavailability and enabling targeted therapy, reducing the frequency of administration and dosage. In tissue engineering, modified PLGA scaffolds can provide better support for cell adhesion, proliferation, and differentiation, with controlled degradation rates that align with the growth of new tissue, facilitating tissue regeneration and repair [60]. Surface-modified PLGA can also reduce inflammatory responses and the formation of immunosuppressive microenvironments, thereby improving the efficacy of immunotherapies.

### 3.3. Surface Modification Strategies for Tumor Immune Regulation and Targeted Delivery

By combining the mechanism of PLGA’s action in the human body, it is not difficult to find that the body’s immunomodulatory effect against PLGA is the biggest challenge for its use in humans.

We delineate PLGA surface modification strategies for immunomodulation and targeted delivery into two distinct categories. The first category pertains to the alteration of the intrinsic properties of PLGA molecules, with a focus on enhancing their immunogenicity and biocompatibility. These modifications are designed to ensure the improved circulation of PLGA as a carrier within the human body and to diminish the risk of premature clearance by the immune system. Optimizing these properties is essential for the longevity and effectiveness of PLGA-based drug delivery systems. The second category involves acquired targeted modifications, where researchers implement specific alterations to the PLGA molecule’s surface to fulfill the target specificity demands for applications such as drug delivery. By adorning the surface of PLGA molecules with precise functional groups or biomolecules, these modifications allow for the targeted delivery of therapeutics to specific cellular or tissue sites. This targeted approach is particularly pertinent in the realm of precision medicine, where the selective presentation of drugs can significantly enhance treatment efficacy and minimize off-target effects [61].

#### 3.3.1. Modification of Intrinsic Properties of PLGA Molecules

R. Yang et al. proposed a strategy to construct cancer vaccines by modifying immunoadjuvant NPs with mannose and encapsulating them with cancer cell membranes. Through the modification of the mannose moiety, the obtained nanovaccine shows enhanced uptake of APCs (e.g., DCs) and then stimulates their maturation state to trigger an anti-tumor immune response [23]. 

This approach is akin to the enhanced cross-presentation induced by protamine-modified PLGA NPs, as previously discussed, which effectively stimulates cytotoxic T lymphocyte (CTL) immune responses. Such stimulation is crucial for the elimination of various infectious diseases and tumors [62].

Similarly, the PEI-coated polymer NPs made by C. Song. could be used as an efficient antigen delivery vehicle and could induce antigen cross-presentation and strong cytotoxic T lymphocyte immune response, to carry out effective anti-cancer immunotherapy [56].

#### 3.3.2. Epigenetic Targeted Modification of PLGA Molecules

Targeted delivery to specific cells or tissues can be accomplished by adorning PLGA NPs with ligands that bind to cell surface receptors. These ligands may include antibodies, oligonucleotides, or other molecular entities that exhibit receptor specificity [63]. For example, Yu et al.’s 1-ethyl-3-(3-dimethylaminopropyl) carbodiimide/N-hydroxysuccinimide -activated biotin-PEG-amine, streptavidin, and biotinylated epithelial cell adhesion molecular antibody (biotin anti-EpCAM)-modified PLGA has a highly efficient ability to capture circulating tumor cells [64].

Furthermore, the fusion of specific cell membrane proteins onto PLGA NPs via surface modification can facilitate targeted delivery by fusing with specific cell membranes. X. Ma et al. have shown that coating PLGA with mature dendritic cell membranes can enhance the targeting ability of dendritic cells, traverse the blood-brain barrier, and induce the maturation of immature dendritic cells, thereby amplifying the activation of immune cells [65].

Additionally, some researchers have employed membranes from cancer cells, such as Lewis lung cancer cells, to camouflage PLGA NPs, which can augment the internalization of NPs and improve the efficacy of drug delivery [66].

Modifications to PLGA that are sensitive to physiological conditions or external stimuli, such as pH-responsive, enzyme-sensitive, or temperature-sensitive, to achieve targeted release in a specific environment [50,53]. 

However, regardless of whether it is unmodified or modified PLGA, it is crucial for researchers and clinicians to consider the material’s interaction with the human immune system to prevent adverse immunological responses. This section elucidates the impact of surface modifications on PLGA’s interaction with the immune system and outlines the considerations and strategies that researchers should employ when designing PLGA NPs for clinical applications.

### 3.4. Synthesis Method of PLGA

In the evaluation and comparison of PLGA synthesis methods, it is beneficial to discuss several key dimensions: ease of operation, particle size control, drug encapsulation efficiency, process complexity, and suitability for large-scale production. Furthermore, the common synthesis methods of PLGA are present in Table 1 and Figure 4.

Firstly, both Solvent Evaporation and Solvent Injection methods are widely used for microparticle preparation. Solvent Evaporation offers better control of particle size, but its main drawbacks include potential solvent residues and high purification requirements. In contrast, Solvent Injection is simpler to operate and suitable for sensitive drugs, but it faces difficulties in controlling particle size and may cause polymer degradation. Nanoprecipitation offers a balance between the two, excelling in nanoparticle preparation and high drug encapsulation efficiency, though it requires careful selection of solvent and cosolvent.

On the other hand, Double Emulsion and Spray Drying cater to specific needs. Double Emulsion is particularly suited for encapsulating water-soluble drugs, despite its complex process and potential for high drug loss. Spray Drying performs excellently in terms of speed and efficiency, making it ideal for mass production, but its high equipment requirements and potential instability with thermosensitive substances limit its application scope. Phase Separation is characterized by its ability to prepare porous materials, although its strict requirements for solvent and condition control increase the complexity of the process.

Advanced techniques such as microfluidics and print method, Electrospinning, and Microfluidic Control represent the forefront of high-precision control. These methods achieve extremely accurate particle size and shape control, particularly suitable for complex formulations. However, they typically require specialized equipment and significant investment costs, and pose challenges in scaling up to industrial levels.

The integration of 3D printing technology with biodegradable polymers like PLGA opens new avenues for the fabrication of complex and patient-specific medical devices. Recent advancements in 3D printing, such as the direct pellet three-dimensional printing of polybutylene adipate-co-terephthalate, highlight the potential for sustainable manufacturing processes [79]. Studies on the 4D printing of porous PLA-TPU structures demonstrate the effect of applied deformation and other factors on the shape memory performance of printed materials, which could be translated to PLGA for advanced biomedical applications [80]. Furthermore, a comprehensive review of various FDM (Fused Deposition Modeling) mechanisms used in the fabrication of continuous-fiber reinforced composites provides insights into how these mechanisms can be adapted for the production of reinforced PLGA structures with enhanced mechanical properties [81]. These research collectively deepen our understanding of how 3D printing can be leveraged to create innovative PLGA-based solutions for immunotherapy and other medical applications.

In summary, although each method has its unique advantages and applications, selecting the appropriate PLGA synthesis method requires balancing factors such as ease of operation, control precision, production scale, and cost-effectiveness. Future research and technological development may focus on enhancing the environmental friendliness of operations, reducing drug loss, decreasing dependency on costly equipment, and improving production output and drug encapsulation efficiency.

### 3.5. Bibliometrics

We use bibliometric methods to analyze the modification of PLGA. We collected 1818 PLGA articles from the Web of Science database by PLGA (Topic) AND modifi* (Topic), selected by criteria published from 2018 to 2024 (Figure 5). The data demonstrate the significance of modification methods in PLGA immunotherapy and reveal various modification approaches and application domains.

## 4. Application of PLGA in Drug Delivery in Tumor Immunotherapy

### 4.1. Encapsulating Traditional Medicines

The therapeutic horizon for a multitude of drugs has been significantly expanded through the drug loading capacity of poly(lactic-co-glycolic acid) (PLGA), offering promising clinical implications. Numerous experimental studies have encapsulated a range of conventional antineoplastic agents within PLGA, including but not limited to paclitaxel, vinorelbine, cisplatin, etoposide, 9-nitrocamptothecin, and amrubicin. These studies have aimed to evaluate the antineoplastic efficacy of the resulting nanoparticles both in vitro and in vivo conditions [74]. The findings suggest that the use of anticancer drug NPs in PLGA may enhance anticancer effects.

Recent studies have demonstrated the principles and effects of using PLGA to load new drugs. This has enabled untested novel drugs, such as laurel oil, and some traditional medicines with initially unclear direct effects, such as many Chinese herbal medicines like Epimedium and Magnolia Officinalis (honokiol), to exhibit a more pronounced antitumor effect following delivery via PLGA [78,82,83]. The potential for drug loading in PLGA to augment the antitumor effects of established medications and to uncover the therapeutic capabilities of traditional medicines remains a fertile ground for scientific inquiry. The encapsulation of these agents in PLGA nanoparticles offers a promising avenue for improving the delivery and efficacy of a broad spectrum of therapeutic compounds.

### 4.2. Encapsulating Enzymes or Other Proteins Targeting Disease

PLGA serves as a versatile carrier not only for the encapsulation of drugs but also for enzymes and specific proteins that are integral to cancer treatment. When untargeted, these bioactive molecules may fail to exert their anti-tumor effects effectively at the lesion site. However, their therapeutic potential can be unlocked through targeted delivery, facilitated by conjugation with PLGA and subsequent surface modification strategies. For instance, the combination of methionine gamma-lyase and pemetrexed inhibits the growth of gastric cancer cells. Methionine gamma-lyase enhances pemetrexed’s inhibitory effects on thymidylate synthase synthesis and cell apoptosis [69]. Additionally, water-soluble catalase could locally generate oxygen, improving the effectiveness of radiotherapy and reducing tumor hypoxia-related radiotherapy resistance [7].Studies have shown that targeting bromodomain-containing protein 4 with PROTACs could be an effective treatment for lung cancer. Overexpression of bromodomain-containing protein 4 is associated with poor prognosis in lung cancer. Inhibiting its expression promotes cell apoptosis and leads to tumor shrinkage [84]. The latest advances in basic research allow for the identification of molecular factors that can either disrupt or bolster specific cellular pathways. This knowledge paves the way for targeted therapeutic interventions using PLGA, offering a promising avenue for the treatment of various diseases with precision and selectivity.

### 4.3. Encapsulating Cytokines

PLGA could also encapsulate small molecules in the body, including inflammatory factors, siRNA, and anti-miRNA. Compared to free IL-10, IL10-NP significantly reduces airway hyperresponsiveness and T-helper 2 /T-helper 17 cell cytokines induced by house dust mite (HDM) in a mouse model and inhibits the increase of neutrophils and eosinophils in the airways, making it a potential treatment for allergic airway diseases [85]. This encapsulation strategy enhances the anti-inflammatory potency of IL-10. In another instance, PLGA encapsulating a complex of soy lecithin and IL-4 has been shown to induce a stable and sustained release of IL-4, which reprograms macrophages within the microenvironment into M2Mφ anti-inflammatory type, thereby inhibiting local excessive inflammatory responses [86]. Encapsulation of transforming growth factor-beta 1 (TGF-β1) within PLGA has also been demonstrated to attenuate immune rejection in the context of allogeneic islet transplantation for type 1 diabetes treatment. By designing a controlled and local release of TGF-β1 for immune regulation and co-cultivating it with naïve CD4 T cells in vitro, it could effectively generate multi-clonal and antigen-specific induced regulatory T cells with strong immune inhibitory function [45]. This body of research suggests that PLGA-encapsulated TGF-β1 could be instrumental in reducing the incidence of immune rejection post-transplantation.

### 4.4. Encapsulating Antigens as Vaccines

PLGA nanoparticles serve as a versatile platform for the encapsulation of therapeutic drugs aimed at targeted delivery to treat various diseases, as well as for the encapsulation of antigens to bolster the immune response, effectively functioning as a vaccine adjuvant [48]. After encapsulating OVA in PLGA, it is effectively internalized in DCs through phagocytosis or macropinocytosis. This process induces efficient cross-presentation of antigens on MHC class I molecules through endosomal escape and lysosomal processing mechanisms, thereby enhancing cellular immunity [56]. Research has shown that the efficiency of cross-presentation of CD8 T cell activation on MHC I molecules by APCs is dependent on the type of polymer used. This significantly increases T cell activation in vitro [48]. In terms of PLGA-encapsulated Angelica sinensis polysaccharide, it has the potential to induce strong and long-lasting humoral and cellular immune responses as a vaccine adjuvant delivery system [55]. ASP is an immune stimulant that acts as an adjuvant in ASP-PLGA-PEI NPs to promote antigen presentation [47]. This leads to the promotion of specific IgG immune responses and cytokine levels, inducing a mixed Th1/Th2 (cellular/humoral) immune response with a Th1 bias [54]. The antigen presentation capabilities of PLGA are particularly noteworthy, underscoring its significance in the establishment of active immunity within the body. These attributes suggest a promising trajectory for the use of PLGA as a component of vaccine formulations.

### 4.5. Encapsulating Contrast Media for Imaging Diagnosis

Cui-Wei Wang and colleagues have studied the application of a new type of ultrasound contrast agent, C3F8, encapsulated in PLGA. They have synthesized and characterized this in the laboratory and conducted in vitro and in vivo studies [87]. C3F8, recognized for its stability, is a gas commonly utilized in the enhancement of ultrasound imaging as a contrast agent. The encapsulation of C3F8 within PLGA nanoparticles is shown to augment the stability and longevity of the agent within the biological milieu. This innovation has the potential to significantly improve the quality and diagnostic accuracy of ultrasound imaging, particularly in the context of breast cancer detection [88].

### 4.6. Encapsulating Compounds for Photothermal and Photodynamic Therapy

In light of challenges such as drug resistance, recent studies have identified novel therapies utilizing PLGA, including photothermal therapy and photodynamic therapy. Photothermal therapy for cancer involves the use of NPs assisted by anti-EGFR antibodies to effectively enter head and neck cancer cells and convert near-infrared light into heat, triggering the release of chemotherapy drugs from the PLGA core and causing tumor ablation through high temperatures [89]. Furthermore, photodynamic therapy utilizes PLGA-encapsulated molybdenum cluster compounds for the treatment of ovarian cancer. Once the inorganic molybdenum octahedral clusters are released from the NPs system, they generate singlet oxygen, leading to reduced cell viability [68]. These emerging therapies underscore the versatility of PLGA and its potential for various research directions based on its physicochemical characteristics. The ability of PLGA to serve as a platform for drug delivery in combination with light-based therapies presents a promising frontier in cancer treatment.

### 4.7. Encapsulating Gene Expression Regulation and Gene Editing Substances

Furthermore, PLGA nanoparticles (NPs) are also being explored for the delivery of small interfering RNA (siRNA) to modulate gene expression, such as using PLGA-encapsulated Stat3siRNA for the treatment of lung cancer [90]. Alternatively, anti-miR-21 could be used for the treatment of triple-negative breast cancer [91]. This approach indicates that there are many possibilities for further exploration of PLGA encapsulation. Ngoc B Nguyen et al. found that PLGA NPs and other nanomaterials could bind to target cells and activate Cre recombinase, leading to tissue specific Cre activation. This system provides a universal and powerful method for inducing recombination in ubiquitous Cre systems for various biomedical applications and lays the foundation for a time- and cost-effective strategy for generating new transgenic mouse strains [92]. These collective studies suggest that PLGA could be harnessed for gene editing or the regulation of gene expression within cells. This includes the direct reprogramming of tumor cells or the reprogramming of other cellular components within the body, such as immune cells, to augment their tumoricidal effects. The potential applications of PLGA in these areas are vast and warrant further investigation.

### 4.8. Problems and Challenges of PLGA in Immunotherapy

Although PLGA encapsulated drugs play a large role in tumor immunotherapy (Table 2), PLGA particles may exhibit immunogenicity and immune stimulation in certain cases, potentially triggering immune and inflammatory responses. This could impact treatment efficacy and limit their widespread clinical application. Therefore, on one hand, modifications could be made to PLGA to alter its inherent immunogenicity and improve its tissue compatibility. On the other hand, the properties of PLGA itself could be utilized, as it possesses certain anti-inflammatory and immunosuppressive properties [20,93]. However, research in this area is limited, and the specific mechanisms are still not fully understood.

Biological factors in the tumor microenvironment could also impact the efficacy of PLGA particles, such as tumor angiogenesis, tumor immune evasion, and others. This means that tumor immunotherapy using PLGA requires a comprehensive consideration of multiple factors, not just focusing on tumor cells themselves from the perspective of cell apoptosis, but also considering the immune aspects of tumor treatment with the stability of PLGA particles in vivo may be influenced by factors such as the accumulation of degradation products and particle aggregation. This could lead to a gradual decline in the functionality and performance of the particles, thereby affecting the therapeutic effects. While the impact of PLGA degradation products and the resulting immune reactions are discussed in the immunological basis of PLGA, studies on the post-degradation effects of PLGA are still insufficient. This hinders a more comprehensive consideration of PLGA in the process of immunotherapy and the improvement of its therapeutic effects. Therefore, there is a broad research prospect for the study of PLGA degradation products.

The rate of drug release from PLGA particles is predominantly governed by the degradation rate of the polymeric material. However, the degradation rate of PLGA is fixed and difficult to adjust during the treatment process. This may result in rapid drug release in the early stages or slow release in the later stages, affecting the therapeutic effects. On one hand, better modifications could be induced to achieve targeted effects, allowing for precise delivery under specific degradation rates. On the other hand, drug targeting or response to physiological conditions could be achieved through external magnetic fields or internal pH endogenous signals [53]. Additionally, the application of photothermal therapy has revealed the possibility of regulating the degradation rate of PLGA through near-infrared means, highlighting the importance of studying the physical properties of PLGA [89]. These developments underscore the importance of research into the physical and chemical properties of PLGA, as they are crucial for the development of smarter, more effective drug delivery systems.

Achieving specific targeting of tumor tissues by PLGA particles in vivo is a complex endeavor that, while improvable, remains challenging. Surface modification and functionalization strategies have shown promise in enhancing the targeting capabilities of PLGA particles. Nonetheless, there are ongoing challenges to address, including the stability of the particles, the drug loading capacity, and the efficiency of targeting. On one hand, it is crucial to deepen our understanding of surface modification techniques and to explore innovative approaches that can lead to improved tissue or organ-specific targeting. This includes the development of biocompatible and biodegradable modifications that not only enhance targeting but also maintain the integrity and bioactivity of the encapsulated therapeutics. Moreover, the optimization of these modifications to overcome current limitations is a critical area of focus. This may involve the design of multifunctional surfaces that can respond to specific biological cues or the incorporation of targeting ligands that have high affinity and specificity for receptors overexpressed on tumor cells. Advancing our knowledge in this field is essential for the development of PLGA-based drug delivery systems that can effectively discriminate between malignant and healthy tissues, thereby maximizing therapeutic outcomes and minimizing side effects.

The drug loading capacity of PLGA particles is limited by their size, morphology, and surface properties. Due to the typically small size of PLGA particles, their drug loading capacity is restricted. Additionally, certain drugs may interact with PLGA, leading to reduced drug loading and release efficiency. Therefore, it is important to study the physical properties of PLGA, analyze the effects of size, morphology, charge, and other physical characteristics on drug loading capacity, and determine the optimal PLGA size, morphology, and charge for achieving the highest drug loading capacity. Furthermore, combining PLGA with other biodegradable polymers could be explored to increase drug loading capacity.

### 4.9. Bibliometrics

We use bibliometric methods to analyze the application of PLGA in the field of cancer immunotherapy. We collected 781 PLGA articles from the Web of Science database on cancer immunotherapy (PLGA (Topic) AND tumor (Topic) AND immun* (Topic)), selected by criteria published from 2018 to 2024, and finally analyzed a total of 521 articles (Figure 6). Bibliometric analysis reveals the significant role of PLGA in tumor immunotherapy, particularly in modulating immune cells such as DCs and CD8+ T cells. Furthermore, it emphasizes the importance of considering the influence of the tumor microenvironment.

## 5. Discussion

In summary, PLGA, as a biodegradable polymer, plays an important role in tumor immunotherapy. It improves the efficacy of immunotherapy by enhancing antigen presentation function, inducing inflammatory inhibition, mediating immune responses, and other pathways. In addition, surface modification of PLGA could reduce its immunorecognition, increase immunocompatibility, and lead to human immunosuppressive regulation, thereby further improving the stability and antigen adjuvant ability of PLGA particles.

PLGA is a versatile drug carrier with promising clinical applications. It has been shown to enhance the efficacy of anticancer drugs, both traditional and novel, when encapsulated in PLGA NPs. Additionally, PLGA could be used to encapsulate enzymes, proteins, and small molecules for targeted cancer therapy and immunomodulation. As a vaccine adjuvant, PLGA enhances the immune response, making it an attractive option for vaccine development. Furthermore, PLGA could be used to encapsulate ultrasound contrast agents, enabling more accurate and reliable imaging for the diagnosis of diseases such as breast cancer. It could also be utilized in novel therapies like photothermal and photodynamic therapy, which have shown potential in cancer treatment. Moreover, PLGA could encapsulate siRNAs and miRNAs, allowing for gene regulation and gene therapy applications. The application of PLGA in gene editing, such as Cre recombinase-mediated recombination, offers a powerful tool for manipulating cellular functions. These highlight the immense potential of PLGA in drug delivery and therapeutic interventions. 

Therefore, based on the discussion of the immunological mechanism of PLGA itself, this review describes the action mechanism of PLGA after modification and the pathogenic mechanism of PLGA after encapsulation of drugs (Figure 7). Our innovation lies in the joint discussion of PLGA with immunological mechanism and immunological therapy, aiming to let researchers understand the immunological effects of PLGA and its research prospects.

In addition to PLGA, there are other biodegradable polymers that also play an important role in tumor immunity. For example, PEG could improve the bioavailability of drugs by improving their solubility and stability. Polyglycolic acid (PLA) and polylactic acid (monomers of PLGA) have good biocompatibility and biodegradability and could be used in sustained-release drug systems [99,100]. Deacetylated chitosan (CS) has good adhesion and biological activity and could be used for targeted drug delivery [101].

The application of PLGA nanoparticles in tumor immunotherapy significantly enhances the efficacy of cancer treatment through various mechanisms. Acting as a drug delivery system, PLGA nanoparticles can encapsulate immunomodulators, anticancer drugs, or vaccines, providing controlled release and increased concentration at the tumor microenvironment, thereby amplifying immune responses and antitumor effects [12,102]. For instance, PLGA nanoparticles can serve as a peptide/protein vaccine delivery system, enhancing immune responses as referenced in literature. Furthermore, by modifying the surface with specific targeting ligands such as antibodies, peptides, or aptamers, PLGA nanoparticles can target tumor cells or receptors within the tumor microenvironment, improving treatment specificity and reducing systemic side effects [84]. In combination therapies, PLGA nanoparticles can co-deliver multiple drugs, such as chemotherapeutics and immunoadjuvants, to achieve synergistic therapeutic effects, activating the immune system while directly targeting cancer cells [23]. Additionally, PLGA nanoparticles themselves or their degradation products can act as immunoadjuvants, activating the immune system and enhancing the antitumor immune response [103,104]. Moreover, PLGA nanoparticles can act as carriers for tumor antigens and immunoadjuvants, promoting the generation of specific immune responses. In the realm of gene therapy, PLGA nanoparticles can be utilized to deliver agents like siRNA or CRISPR systems to silence key genes in cancer cells or modulate immune checkpoints, increasing the sensitivity of tumor cells to immune attacks [63]. Lastly, PLGA nanoparticles can deliver drugs that modulate the tumor microenvironment, such as anti-angiogenic factors or immunomodulators, facilitating immune cell infiltration and tumor cell elimination [105]. The potential applications of PLGA nanoparticles in tumor immunotherapy are broad and promising, offering more effective and safer treatment options for cancer patients. However, to realize these potential applications, further optimization of nanoparticle design, fabrication processes, and administration strategies is necessary, along with clinical trials to verify their safety and efficacy.

The versatility of PLGA as a drug carrier significantly contributes to its potential clinical applications in cancer therapy, particularly through targeted therapy, controlled release, and combination treatments [70,106,107]. Compared to other biodegradable polymers, PLGA offers unique advantages such as targeted delivery through surface modification, adjustable drug release rates by altering copolymer ratios, and enhanced patient compliance with reduced injection frequency. PEG, while biocompatible and stable, lacks biodegradability, which may lead to long-term accumulation and potential toxicity [108]. PLA has a slower degradation rate, making it less suitable for therapies requiring rapid drug release [100]. CS, despite its biocompatibility and immunoadjuvant properties, may not match PLGA in terms of degradation rate and drug encapsulation efficiency [109]. Challenges associated with the application of biodegradable polymers in cancer therapy and other fields include understanding their biodistribution and clearance, precise control of drug release rates, enhancing targeting and biocompatibility, and scaling up production for clinical use [110]. Future research efforts should focus on in-depth studies of the biological effects of these polymers, development of novel polymers and nanoparticle designs, optimization of surface modifications for improved targeting and biocompatibility, and conducting preclinical and clinical studies to verify the safety and efficacy of new drug delivery systems, thereby overcoming current challenges and advancing the clinical application of biodegradable polymers in cancer therapy and other biomedical fields.

In addition to tumor immunotherapy, PLGA is also widely used in other fields. For example, it could be used in tissue engineering to repair and regenerate damaged tissue [111]. In addition, PLGA could also be used for bone regeneration, promoting the growth and repair of bone tissue through drug loading and sustained-release drugs [112]. It is also widely used in the treatment of other diseases. PLGA could be used in many fields such as cardiovascular disease treatment, orthopedic treatment, neurological disease treatment, and infectious disease treatment [113]. In the treatment of cardiovascular diseases, PLGA could be used to prepare vascular stents, drug release systems, and cardiac repair materials to promote vascular regeneration and cardiac function recovery [114]. In orthopedic treatment, PLGA could be used to prepare bone repair materials and bone scaffolds to promote bone regeneration and bone defect repair. In the treatment of neurological diseases, PLGA could be used to prepare nerve repair materials and nerve regeneration catheters to promote nerve regeneration and neurological recovery. In the treatment of infectious diseases, PLGA could be used to prepare antimicrobial sustained-release systems and vaccine delivery systems to improve treatment efficacy and prevent infection [115]. In general, PLGA, as a multifunctional biodegradable polymer, has a wide range of application prospects in the treatment of multiple diseases.

In summary, PLGA, alongside other biodegradable polymers, occupies a significant role in the realm of tumor immunology and presents a broad spectrum of application prospects across various fields. The utility of these polymers in drug delivery and therapeutic applications is particularly noteworthy. Despite their potential, the application of biodegradable polymers like PLGA is not without challenges. Current limitations include difficulties in achieving a controlled drug release rate and ensuring sufficient specificity in targeting. These hurdles necessitate ongoing research and innovation to refine the properties and performance of these materials. Further advancements are essential to enhance the efficacy of biodegradable polymers in tumor immunity and to expand their applications in other domains. This includes the development of more sophisticated targeting strategies, improvement in the controllability of drug release kinetics, and the exploration of novel functionalization techniques to tailor the polymers to specific therapeutic needs. Addressing these challenges will not only optimize the use of PLGA in cancer treatment but also unlock new possibilities for the treatment of other diseases, thereby contributing to the broader landscape of biomedical research and clinical practice.

## Figures and Tables

**Figure 1 polymers-16-01253-f001:**
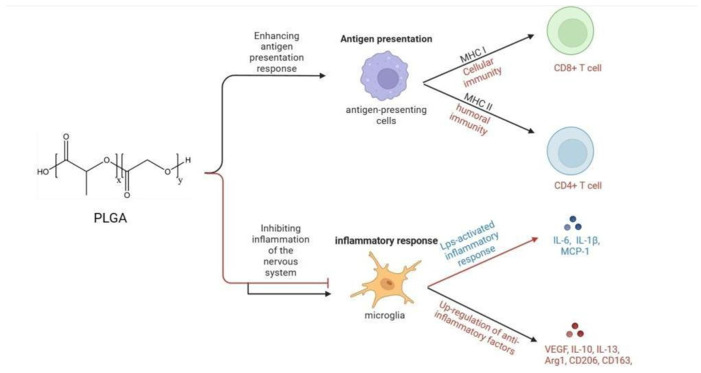
Immunological mechanisms of PLGA. PLGA enhances humoral and cellular immunity by affecting the antigen presentation process, and inhibits inflammation of the nervous system by inhibiting the secretion of anti-inflammatory factors and promoting the synthesis of anti-inflammatory factors by microglia.

**Figure 2 polymers-16-01253-f002:**
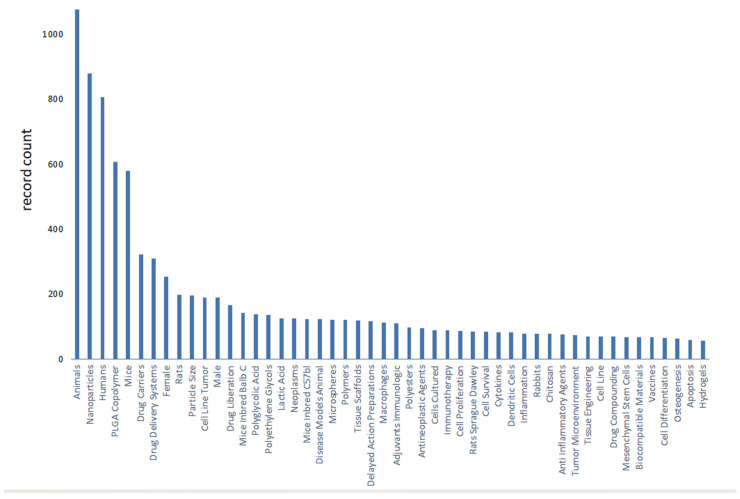
Bibliometric analysis about PLGA and immunologic mechanism. It presents immune cells, cytokine and inflammation such as dendritic cell and macrophages. It also shows the importance of immunological mechanism of PLGA in the field of studies on tumor immune microenvironment.

**Figure 3 polymers-16-01253-f003:**
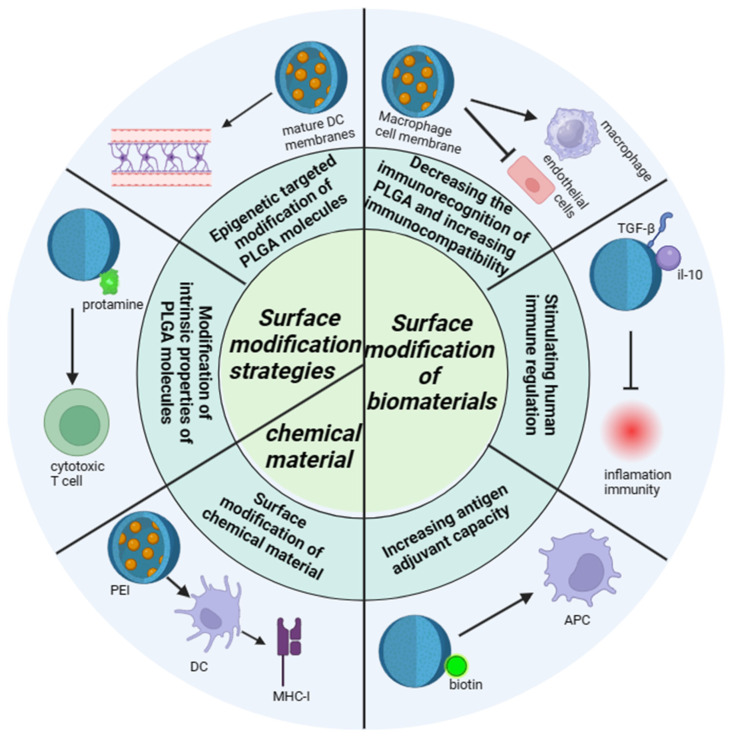
The modification mechanism of PLGA. The modification mechanism figure of PLGA shows that PLGA affects inflammatory response and immune effect through interaction with immune cells after modification of biological materials and chemical materials, as well as the barrier passing ability of PLGA influenced by modification strategies; a normal arrow means facilitation or activation, and an arrow with a crossbar means inhibition.

**Figure 4 polymers-16-01253-f004:**
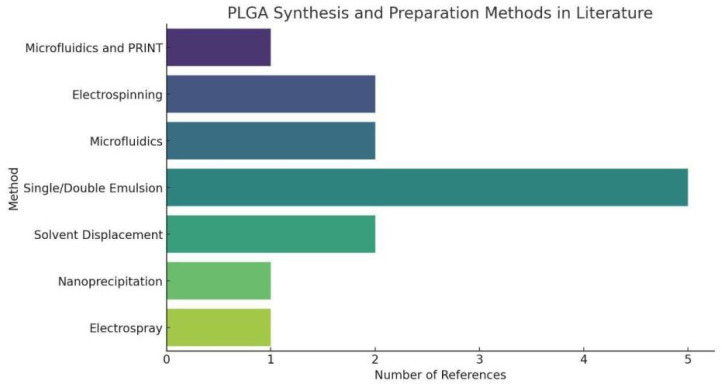
PLGA Synthesis and Preparation Methods in Literature. The statistical chart of PLGA Synthesis and Preparation Methods simply shows that Single/Double Emulsion has a large proportion in PLGA synthesis, indicating the universality of this PLGA synthesis method.

**Figure 5 polymers-16-01253-f005:**
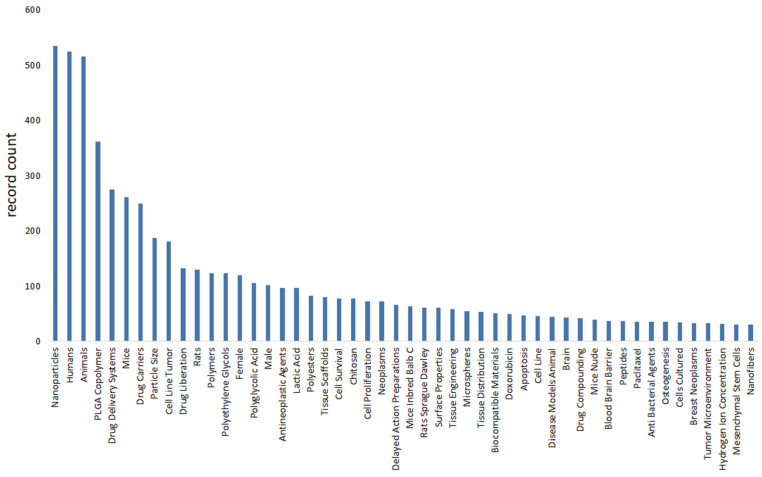
The bibliometrics analysis of literature on PLGA and its modification through web of science. It shows the importance of modification methods in PLGA immunotherapy, and reveals various modification methods and application fields, such as peptides, chitosan and Doxorubicin.

**Figure 6 polymers-16-01253-f006:**
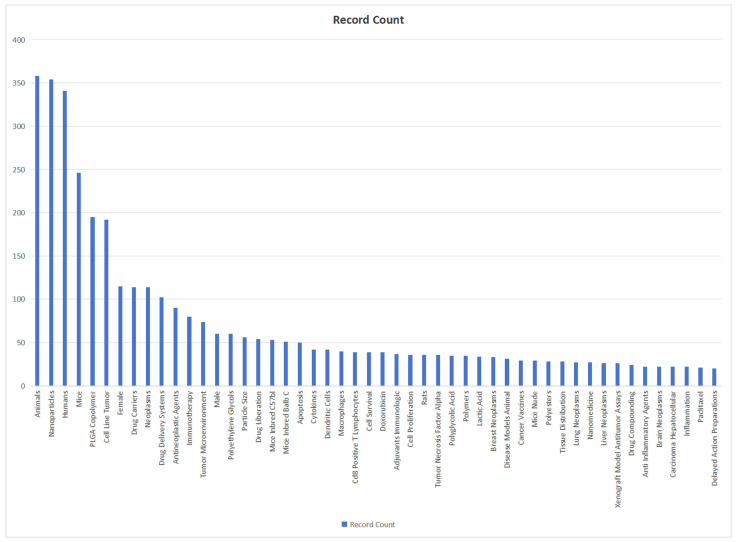
Bibliometric analysis about PLGA and tumor immunology. It reveals that the hot spot of PLGA tumor therapy lies in the regulation of immune cells such as DC and CD8+ T cells, which emphasizes the importance of tumor microenvironment.

**Figure 7 polymers-16-01253-f007:**
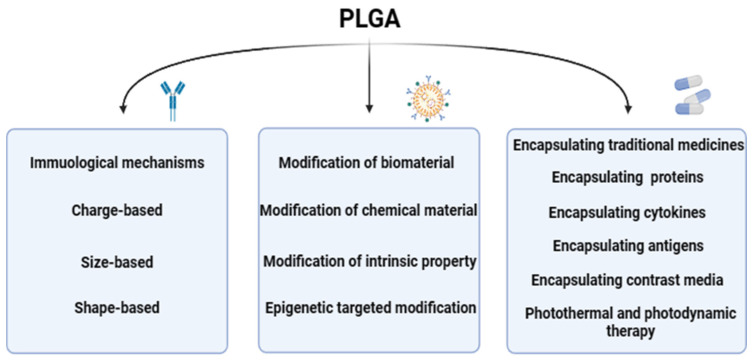
An overview of the article. It presents the effect of charge size and shape of PLGA on its immunological mechanism, the mechanism of modification by different biological or chemical materials, and the treatment methods presented after encapsulation of different drugs and new therapeutic methods.

**Table 1 polymers-16-01253-t001:** PLGA modification methods and their advantages and disadvantages.

Preparation Method	Advantages	Disadvantages	Reference
Solvent Evaporation	Commonly used for microparticle preparation, good control of particle size	Possible residual solvents, high purification required	[67,68,69]
Solvent Injection	Fast and easy to operate, suitable for sensitive drugs	Difficult control of particle size, possible polymer degradation	[70]
Nanoprecipitation	Suitable for nanoparticles, good drug encapsulation efficiency	High requirements for solvent and cosolvent selection	[71]
Double Emulsion	Suitable for encapsulating water-soluble drugs	Complex preparation process, potential high drug loss	[69,72,73,74]
Spray Drying	Fast and efficient, suitable for mass production	High equipment requirements, potential instability of thermosensitive substances	[14]
Phase Separation	Suitable for the preparation of porous materials	Strict requirements for solvent and condition control	[75,76]
Microfluidics and PRINT Method	High precision in particle size and shape control, suitable for complex formulations	Requires specialized equipment and expertise, high setup cost	[77]
Electrospinning	Produces fibers with high surface area to volume ratio, versatile in creating structures	Requires high voltage, difficult to scale up for industrial production	[64,78]
Microfluidic Control	Enables precise control of fluid flow and mixing at microscale, suitable for uniform particle production	Complexity in design and fabrication, scalability issues	[14,64]

**Table 2 polymers-16-01253-t002:** Encapsulation, immune mechanism and effect of PLGA for disease treatment.

Drug Encapsulation	Effectiveness	Targeted Immune Cell Type	Targeted Disease	Reference
CREATE (nano-PROTAC)	Induces significant apoptosis in lung cancer cells and M2 macrophages	Lung cancer cells and M2 macrophages	Lung Cancer	[84]
Cabozantinib -loaded PLGA	Promotes macrophage polarization	Macrophages	Cancer	[94]
PLGA Nanoparticles	Successfully inhibits miR-155	HeLa cells and M1 macrophages	Cancer	[95]
HA-NPs-17AAG	Induces better apoptosis than 17AAG alone	Not specified	Cancer	[52]
PLGA Microspheres	Effectively releases As_2_O_3_ and HCPT over 10 and 12 days, respectively	Not specified	Cancer	[96]
DNA vaccine targeting FGL1 and CAIX	Delivered via PLGA/PLA nanoparticles	Not specified	Cancer	[97]
TH-302 NPs	Enhances the efficacy of α-PD-1	Immune checkpoint cells	Cancer	[46]
PRECIOUS-01 Immunomodulatory Nanomedicine	Based on PLGA	Not specified	Cancer	[98]

## Data Availability

The data used in our bibliometrics are all derived from results retrieved in web of science in April 2024.

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
