# Peer review of "Application of PLGA in Tumor Immunotherapy"

_polymers, 2024, doi:10.3390/polym16091253_

Round 1
Reviewer 1 Report
Comments and Suggestions for Authors
Application of PLGA in tumor immunotherapy
This article says PLGA, a biodegradable polymer, is extensively researched in biomedicine, particularly in drug delivery systems. It shows promise in various medical applications, including vaccines, tissue engineering, and 3D printing. Recently, it has gained attention in tumor immunotherapy due to its ability to serve as a carrier for antigens, formulation of tumor vaccines, and enhancement of immunoadjuvants. PLGA nanoparticles enhance tumor immunotherapy by regulating immune cell activity and improving tumor antigen expression. Its diverse physical properties and surface modifications offer potential for a wide range of applications in tumor immunotherapy by loading various drugs or innovative substances. The manuscript is well structured but the following should be considered before publishing.
COMMENTS:
1. How does the use of PLGA in tumor immunotherapy enhance the efficacy of cancer treatment, and what are some potential applications of PLGA nanoparticles in this field?
2. How does the use of PLGA in cancer therapy address the limitations of conventional cancer treatments, and what are some specific advantages of using PLGA in drug delivery systems and immunotherapy enhancement?
3. To what extent does understanding the immunological mechanisms of PLGA NPs contribute to optimizing their effectiveness in immunotherapy?
4. Use the following references to deepen the 3D printing section. Direct Pellet Three-Dimensional Printing of Polybutylene Adipate-co-Terephthalate for a Greener Future. 4D printing of porous PLA-TPU structures: effect of applied deformation, loading mode and infill pattern on the shape memory performance. Various FDM Mechanisms Used in the Fabrication of Continuous-Fiber Reinforced Composites: A Review.
5. How do the interactions between PLGA NPs and immune cells influence both short-term and long-term immunological responses, and how can this knowledge be leveraged to enhance the design and clinical application of PLGA-based immunotherapy strategies?
6. How do different surface modification techniques of PLGA, such as nanotechnology modification and biodegradable polymer modification, affect its physicochemical properties and suitability for various biomedical applications, including drug delivery and tissue engineering?
7. How does the versatility of PLGA as a drug carrier contribute to its potential clinical applications in cancer therapy? Additionally, how do other biodegradable polymers, such as PEG, PLA, and CS, compare to PLGA in terms of their suitability for tumor immunotherapy and other biomedical applications? Finally, what are some of the challenges associated with the application of biodegradable polymers in cancer therapy and other fields, and what research efforts are needed to address these challenges?
Comments on the Quality of English Language***
Author Response
Response to Reviewer 1 Comments
- Summary
Thank you for your letter and the reviewers’ comments concerning our manuscript entitled “Application of PLGA in tumor immunotherapy”. Those comments are valuable and very helpful. We have read through comments carefully and have made corrections. Based on the instructions provided in your letter, we uploaded the file of the revised manuscript.
- Point-by-point response to Comments and Suggestions for Authors
Comments 1. How does the use of PLGA in tumor immunotherapy enhance the efficacy of cancer treatment, and what are some potential applications of PLGA nanoparticles in this field?
Response:Thank you for your comments, the discussion regarding this question is presented following: The application of PLGA nanoparticles in tumor immunotherapy significantly enhances the efficacy of cancer treatment through various mechanisms. Acting as a drug delivery system, PLGA nanoparticles can encapsulate immunomodulators, anticancer drugs, or vaccines, providing controlled release and increased concentration at the tumor microenvironment, thereby amplifying immune responses and antitumor effects [1,2]. For instance, PLGA nanoparticles can serve as a peptide/protein vaccine delivery system, enhancing immune responses as referenced in literature. Furthermore, by modifying the surface with specific targeting ligands such as antibodies, peptides, or aptamers, PLGA nanoparticles can target tumor cells or receptors within the tumor microenvironment, improving treatment specificity and reducing systemic side effects [3]. In combination therapies, PLGA nanoparticles can co-deliver multiple drugs, such as chemotherapeutics and immunoadjuvants, to achieve synergistic therapeutic effects, activating the immune system while directly targeting cancer cells [4]. Additionally, PLGA nanoparticles themselves or their degradation products can act as immunoadjuvants, activating the immune system and enhancing the antitumor immune response [5,6]. Moreover, PLGA nanoparticles can act as carriers for tumor antigens and immunoadjuvants, promoting the generation of specific immune responses. In the realm of gene therapy, PLGA nanoparticles can be utilized to deliver agents like siRNA or CRISPR systems to silence key genes in cancer cells or modulate immune checkpoints, increasing the sensitivity of tumor cells to immune attacks [7]. Lastly, PLGA nanoparticles can deliver drugs that modulate the tumor microenvironment, such as anti-angiogenic factors or immunomodulators, facilitating immune cell infiltration and tumor cell elimination [8]. The potential applications of PLGA nanoparticles in tumor immunotherapy are broad and promising, offering more effective and safer treatment options for cancer patients. However, to realize these potential applications, further optimization of nanoparticle design, fabrication processes, and administration strategies is necessary, along with clinical trials to verify their safety and efficacy.
We have added the information required as explained above (Part5 Discussion, Lines 806-832, page 22,23).
Comments 2. How does the use of PLGA in cancer therapy address the limitations of conventional cancer treatments, and what are some specific advantages of using PLGA in drug delivery systems and immunotherapy enhancement?
Response:Thank you for your comment, and our reply is as follows: The use of PLGA nanoparticles in cancer therapy addresses several limitations of conventional treatments such as chemotherapy, which often suffer from uneven drug distribution, severe side effects, and rapid drug elimination. PLGA nanoparticles provide a controlled and sustained release mechanism, reducing toxicity to normal tissues while increasing drug concentration at the tumor site, thus enhancing treatment efficacy [9]. PLGA nanoparticles are biocompatible and biodegradable, with degradation products that can be naturally metabolized, reducing the risk of long-term toxicity. Additionally, the ratio of lactic to glycolic acid in PLGA can be adjusted to tailor the drug release rate for personalized treatment [10]. As a carrier for immunotherapeutic agents, PLGA nanoparticles can simulate the effect of multiple injections through pulsatile release, enhancing immune responses, inhibiting tumor growth, and prolonging survival without the need for repeated injections, thereby improving patient compliance and reducing the risk of treatment-associated metastasis [11]. Surface modification with targeting ligands, such as antibodies or peptides, allows for specific targeting of tumor cells, enhancing treatment efficacy and reducing systemic side effects [12]. The biocompatibility and biodegradability of PLGA nanoparticles ensure their safety in the body, with studies showing complete degradation and clearance inflammation or other toxic reactions [13]. These advantages position PLGA nanoparticles as a promising approach in cancer therapy, offering patients more effective, safer, and more convenient treatment options.
We have added the information required as explained above (Part1 Introduction, Lines 78-96, page 2).
Comments 3. To what extent does understanding the immunological mechanisms of PLGA NPs contribute to optimizing their effectiveness in immunotherapy?
Response:Thank you for your comments, the discussion regarding this question is presented following: Understanding the immunological mechanisms of PLGA nanoparticles is crucial for optimizing their effectiveness in immunotherapy. The immunological mechanisms include the interaction of PLGA nanoparticles with antigen-presenting cells (APCs), their role in enhancing antigen presentation, and their ability to modulate immune responses. By elucidating these mechanisms, the design of PLGA nanoparticles can be tailored to improve their biocompatibility, targeting, and immune-stimulating properties. For example, surface modifications can enhance the interaction of nanoparticles with specific immune cells, thereby improving their therapeutic efficacy. Additionally, understanding the degradation products of PLGA and their impact on the immune system allows for the development of strategies to mitigate potential adverse immunological responses, ensuring the safety and effectiveness of PLGA nanoparticles in clinical applications. This knowledge can guide the design of multimodal nanoparticles capable of drug delivery, immune activation, and modulation of the tumor microenvironment, thereby enhancing the overall efficacy of immunotherapy.
We have added the information required as explained above (Part2 Immunological basis of PLGA in immunotherapy, Lines 165-178, page 4).
Comments 4. Use the following references to deepen the 3D printing section. Direct Pellet Three-Dimensional Printing of Polybutylene Adipate-co-Terephthalate for a Greener Future. 4D printing of porous PLA-TPU structures: effect of applied deformation, loading mode and infill pattern on the shape memory performance. Various FDM Mechanisms Used in the Fabrication of Continuous-Fiber Reinforced Composites: A Review.
Response:We are grateful for the suggestion. To be clearer and in accordance with the reviewer concerns, we have added a brief description as follows: The integration of 3D printing technology with biodegradable polymers like PLGA opens new avenues for the fabrication of complex and patient-specific medical devices. Recent advancements in 3D printing, such as the direct pellet three-dimensional printing of polybutylene adipate-co-terephthalate, highlight the potential for sustainable manufacturing processes [14]. Studies on the 4D printing of porous PLA-TPU structures demonstrate the effect of applied deformation and other factors on the shape memory performance of printed materials, which could be translated to PLGA for advanced biomedical applications [15]. Furthermore, a comprehensive review of various FDM (Fused Deposition Modeling) mechanisms used in the fabrication of continuous-fiber reinforced composites provides insights into how these mechanisms can be adapted for the production of reinforced PLGA structures with enhanced mechanical properties [16]. These research collectively deepen our understanding of how 3D printing can be leveraged to create innovative PLGA-based solutions for immunotherapy and other medical applications.
We have added the information required as explained above (Part3.4 Synthesis method of PLGA, Lines 543-555, page 14).
Comments 5. How do the interactions between PLGA NPs and immune cells influence both short-term and long-term immunological responses, and how can this knowledge be leveraged to enhance the design and clinical application of PLGA-based immunotherapy strategies?
Response: Thank you for your comment, and our reply is as follows: The interactions between PLGA nanoparticles and immune cells significantly influence both short-term and long-term immunological responses, and understanding these interactions is vital for enhancing the design and clinical application of PLGA-based immunotherapy strategies. In the short term, PLGA NPs can enhance antigen presentation, rapidly activating immune cells such as dendritic cells and T cells [17]. Additionally, immunoadjuvants like CpG oligonucleotides, when delivered via PLGA NPs, can quickly activate natural killer cells and T cells, initiating a swift immune response [4]. Over the long term, PLGA NPs can promote the generation of memory T cells, providing lasting immune protection against tumor reoccurrence [18]. Moreover, well-designed PLGA NPs can prevent immune tolerance to therapeutic antigens, ensuring the longevity of the immune response [19]. This knowledge can be utilized to strengthen PLGA-based immunotherapy strategies through targeted delivery, combination therapies, and sustained release systems [20-22]. In clinical applications, personalized treatments, postoperative therapies, and safety assessments are key directions for realizing the potential of PLGA NPs, enhancing the effectiveness of cancer treatment [23,24].
We have added the information required as explained above (Part2 Immunological basis of PLGA in immunotherapy, Lines 150-164, page 4).
Comments 6. How do different surface modification techniques of PLGA, such as nanotechnology modification and biodegradable polymer modification, affect its physicochemical properties and suitability for various biomedical applications, including drug delivery and tissue engineering?
Response: We deeply appreciate the reviewer’s suggestion. According to the reviewer’s comment, we have added a more detailed interpretation regarding surface modification. On the whole,different surface modification techniques of PLGA, including nanotechnology and biodegradable polymer modifications, significantly impact its physicochemical properties and suitability across various biomedical applications. Nanotechnology modifications, such as controlling the size and shape of PLGA at the nanoscale, can enhance hydrophilicity, reduce nonspecific protein adsorption and cell adhesion, thereby improving the circulation time and drug delivery efficiency of PLGA nanoparticles in the body [25]. Surface modifications with specific ligands or antibodies enable targeted delivery to particular cells or tissues, enhancing treatment specificity and reducing side effects [26]. For instance, aptamer-conjugated PLGA nanoparticles showcase the potential of ligand modification for targeted cancer therapy. Biodegradable polymer modifications, such as the covalent attachment of polyethylene glycol (PEG), can create a "stealth surface" on PLGA microspheres, reducing immune recognition and clearance, and prolonging their half-life in the body [27]. These modifications can also adjust the degradation rate of PLGA and the kinetics of drug release to suit different therapeutic needs. For example, altering the ratio of lactic to glycolic acid in PLGA copolymers can tailor the degradation rate and drug release profile[28]. These modifications positively affect drug delivery by enhancing bioavailability and enabling targeted therapy, reducing the frequency of administration and dosage. In tissue engineering, modified PLGA scaffolds can provide better support for cell adhesion, proliferation, and differentiation, with controlled degradation rates that align with the growth of new tissue, facilitating tissue regeneration and repair [29]. Surface-modified PLGA can also reduce inflammatory responses and the formation of immunosuppressive microenvironments, thereby improving the efficacy of immunotherapies.
We have added the information required as explained above (Part3.2. Surface modification of chemical material, Lines 430-451, page 11).
Comments 7. How does the versatility of PLGA as a drug carrier contribute to its potential clinical applications in cancer therapy? Additionally, how do other biodegradable polymers, such as PEG, PLA, and CS, compare to PLGA in terms of their suitability for tumor immunotherapy and other biomedical applications? Finally, what are some of the challenges associated with the application of biodegradable polymers in cancer therapy and other fields, and what research efforts are needed to address these challenges?
Response: Thank you for the suggestion. We have added the information required as explained above (Part5 Discussion, Lines 834-852, page 23). The versatility of PLGA as a drug carrier significantly contributes to its potential clinical applications in cancer therapy, particularly through targeted therapy, controlled release, and combination treatments [30-32]. Compared to other biodegradable polymers, PLGA offers unique advantages such as targeted delivery through surface modification, adjustable drug release rates by altering copolymer ratios, and enhanced patient compliance with reduced injection frequency. PEG, while biocompatible and stable, lacks biodegradability, which may lead to long-term accumulation and potential toxicity [33]. PLA has a slower degradation rate, making it less suitable for therapies requiring rapid drug release [34]. CS, despite its biocompatibility and immunoadjuvant properties, may not match PLGA in terms of degradation rate and drug encapsulation efficiency [35]. Challenges associated with the application of biodegradable polymers in cancer therapy and other fields include understanding their biodistribution and clearance, precise control of drug release rates, enhancing targeting and biocompatibility, and scaling up production for clinical use [36]. Future research efforts should focus on in-depth studies of the biological effects of these polymers, development of novel polymers and nanoparticle designs, optimization of surface modifications for improved targeting and biocompatibility, and conducting preclinical and clinical studies to verify the safety and efficacy of new drug delivery systems, thereby overcoming current challenges and advancing the clinical application of biodegradable polymers in cancer therapy and other biomedical fields.
Thank you for your precious comments and advice. Those comments are all valuable and very helpful for revising and improving our paper. We have revised the manuscript accordingly, and our point-by-point responses are presented above.
- Allahyari, M.; Mohit, E. Peptide/protein vaccine delivery system based on PLGA particles. Hum Vaccin Immunother 2016, 12, 806-828, doi:10.1080/21645515.2015.1102804.
- Narmani, A.; Jahedi, R.; Bakhshian-Dehkordi, E.; Ganji, S.; Nemati, M.; Ghahramani-Asl, R.; Moloudi, K.; Hosseini, S.M.; Bagheri, H.; Kesharwani, P.; et al. Biomedical applications of PLGA nanoparticles in nanomedicine: advances in drug delivery systems and cancer therapy. Expert Opin Drug Deliv 2023, 20, 937-954, doi:10.1080/17425247.2023.2223941.
- Zhang, H.T.; Peng, R.; Chen, S.; Shen, A.; Zhao, L.; Tang, W.; Wang, X.H.; Li, Z.Y.; Zha, Z.G.; Yi, M.; Zhang, L. Versatile Nano-PROTAC-Induced Epigenetic Reader Degradation for Efficient Lung Cancer Therapy. Adv Sci (Weinh) 2022, 9, e2202039, doi:10.1002/advs.202202039.
- Yang, R.; Xu, J.; Xu, L.; Sun, X.; Chen, Q.; Zhao, Y.; Peng, R.; Liu, Z. Cancer Cell Membrane-Coated Adjuvant Nanoparticles with Mannose Modification for Effective Anticancer Vaccination. ACS Nano 2018, 12, 5121-5129, doi:10.1021/acsnano.7b09041.
- Yang, Y.; Teng, Z.; Lu, Y.; Luo, X.; Mu, S.; Ru, J.; Zhao, X.; Guo, H.; Ran, X.; Wen, X.; Sun, S. Enhanced immunogenicity of foot and mouth disease DNA vaccine delivered by PLGA nanoparticles combined with cytokine adjuvants. Res Vet Sci 2021, 136, 89-96, doi:10.1016/j.rvsc.2021.02.010.
- Kabiri, M.; Sankian, M.; Sadri, K.; Tafaghodi, M. Robust mucosal and systemic responses against HTLV-1 by delivery of multi-epitope vaccine in PLGA nanoparticles. Eur J Pharm Biopharm 2018, 133, 321-330, doi:10.1016/j.ejpb.2018.11.003.
- Hashemi, M.; Shamshiri, A.; Saeedi, M.; Tayebi, L.; Yazdian-Robati, R. Aptamer-conjugated PLGA nanoparticles for delivery and imaging of cancer therapeutic drugs. Arch Biochem Biophys 2020, 691, 108485, doi:10.1016/j.abb.2020.108485.
- Yang, M.; Li, J.; Gu, P.; Fan, X. The application of nanoparticles in cancer immunotherapy: Targeting tumor microenvironment. Bioact Mater 2021, 6, 1973-1987, doi:10.1016/j.bioactmat.2020.12.010.
- He, P.; Xu, S.; Guo, Z.; Yuan, P.; Liu, Y.; Chen, Y.; Zhang, T.; Que, Y.; Hu, Y. Pharmacodynamics and pharmacokinetics of PLGA-based doxorubicin-loaded implants for tumor therapy. Drug Deliv 2022, 29, 478-488, doi:10.1080/10717544.2022.2032878.
- Su, Y.; Zhang, B.; Sun, R.; Liu, W.; Zhu, Q.; Zhang, X.; Wang, R.; Chen, C. PLGA-based biodegradable microspheres in drug delivery: recent advances in research and application. Drug Deliv 2021, 28, 1397-1418, doi:10.1080/10717544.2021.1938756.
- Lu, X.; Miao, L.; Gao, W.; Chen, Z.; McHugh, K.J.; Sun, Y.; Tochka, Z.; Tomasic, S.; Sadtler, K.; Hyacinthe, A.; et al. Engineered PLGA microparticles for long-term, pulsatile release of STING agonist for cancer immunotherapy. Sci Transl Med 2020, 12, doi:10.1126/scitranslmed.aaz6606.
- Zhong, Y.; Su, T.; Shi, Q.; Feng, Y.; Tao, Z.; Huang, Q.; Li, L.; Hu, L.; Li, S.; Tan, H.; et al. Co-Administration Of iRGD Enhances Tumor-Targeted Delivery And Anti-Tumor Effects Of Paclitaxel-Loaded PLGA Nanoparticles For Colorectal Cancer Treatment. Int J Nanomedicine 2019, 14, 8543-8560, doi:10.2147/ijn.S219820.
- Danhier, F.; Ansorena, E.; Silva, J.M.; Coco, R.; Le Breton, A.; Préat, V. PLGA-based nanoparticles: an overview of biomedical applications. J Control Release 2012, 161, 505-522, doi:10.1016/j.jconrel.2012.01.043.
- Karimi, A.; Rahmatabadi, D.; Baghani, M. Direct Pellet Three-Dimensional Printing of Polybutylene Adipate-co-Terephthalate for a Greener Future. Polymers (Basel) 2024, 16, doi:10.3390/polym16020267.
- Rahmatabadi, D.; Soltanmohammadi, K.; Aberoumand, M.; Soleyman, E.; Ghasemi, I.; Baniassadi, M.; Abrinia, K.; Bodaghi, M.; Baghani, M. 4D printing of porous PLA-TPU structures: effect of applied deformation, loading mode and infill pattern on the shape memory performance. Physica Scripta 2024, 99, doi:10.1088/1402-4896/ad1957.
- Karimi, A.; Rahmatabadi, D.; Baghani, M. Various FDM Mechanisms Used in the Fabrication of Continuous-Fiber Reinforced Composites: A Review. Polymers (Basel) 2024, 16, doi:10.3390/polym16060831.
- Liu, J.; Liu, X.; Han, Y.; Zhang, J.; Liu, D.; Ma, G.; Li, C.; Liu, L.; Kong, D. Nanovaccine Incorporated with Hydroxychloroquine Enhances Antigen Cross-Presentation and Promotes Antitumor Immune Responses. ACS Appl Mater Interfaces 2018, 10, 30983-30993, doi:10.1021/acsami.8b09348.
- Demento, S.L.; Cui, W.; Criscione, J.M.; Stern, E.; Tulipan, J.; Kaech, S.M.; Fahmy, T.M. Role of sustained antigen release from nanoparticle vaccines in shaping the T cell memory phenotype. Biomaterials 2012, 33, 4957-4964, doi:10.1016/j.biomaterials.2012.03.041.
- Kim, S.H.; Moon, J.H.; Jeong, S.U.; Jung, H.H.; Park, C.S.; Hwang, B.Y.; Lee, C.K. Induction of antigen-specific immune tolerance using biodegradable nanoparticles containing antigen and dexamethasone. Int J Nanomedicine 2019, 14, 5229-5242, doi:10.2147/ijn.S210546.
- Freitas, R.; Ferreira, E.; Miranda, A.; Ferreira, D.; Relvas-Santos, M.; Castro, F.; Santos, B.; Gonçalves, M.; Quintas, S.; Peixoto, A.; et al. Targeted and Self-Adjuvated Nanoglycovaccine Candidate for Cancer Immunotherapy. ACS Nano 2024, 18, 10088-10103, doi:10.1021/acsnano.3c12487.
- Xiao, Q.; Li, X.; Li, Y.; Wu, Z.; Xu, C.; Chen, Z.; He, W. Biological drug and drug delivery-mediated immunotherapy. Acta Pharm Sin B 2021, 11, 941-960, doi:10.1016/j.apsb.2020.12.018.
- Ray, S.; Puente, A.; Steinmetz, N.F.; Pokorski, J.K. Recent advancements in single dose slow-release devices for prophylactic vaccines. Wiley Interdiscip Rev Nanomed Nanobiotechnol 2023, 15, e1832, doi:10.1002/wnan.1832.
- Zhang, Y.; Wang, T.; Tian, Y.; Zhang, C.; Ge, K.; Zhang, J.; Chang, J.; Wang, H. Gold nanorods-mediated efficient synergistic immunotherapy for detection and inhibition of postoperative tumor recurrence. Acta Pharm Sin B 2021, 11, 1978-1992, doi:10.1016/j.apsb.2021.03.035.
- Tang, X.D.; Lü, K.L.; Yu, J.; Du, H.J.; Fan, C.Q.; Chen, L. In vitro and in vivo evaluation of DC-targeting PLGA nanoparticles encapsulating heparanase CD4(+) and CD8(+) T-cell epitopes for cancer immunotherapy. Cancer Immunol Immunother 2022, 71, 2969-2983, doi:10.1007/s00262-022-03209-1.
- Rocha, C.V.; Gonçalves, V.; da Silva, M.C.; Bañobre-López, M.; Gallo, J. PLGA-Based Composites for Various Biomedical Applications. Int J Mol Sci 2022, 23, doi:10.3390/ijms23042034.
- Zhang, D.; Liu, L.; Wang, J.; Zhang, H.; Zhang, Z.; Xing, G.; Wang, X.; Liu, M. Drug-loaded PEG-PLGA nanoparticles for cancer treatment. Front Pharmacol 2022, 13, 990505, doi:10.3389/fphar.2022.990505.
- Rafiei, P.; Haddadi, A. Docetaxel-loaded PLGA and PLGA-PEG nanoparticles for intravenous application: pharmacokinetics and biodistribution profile. Int J Nanomedicine 2017, 12, 935-947, doi:10.2147/ijn.S121881.
- Hua, Y.; Su, Y.; Zhang, H.; Liu, N.; Wang, Z.; Gao, X.; Gao, J.; Zheng, A. Poly(lactic-co-glycolic acid) microsphere production based on quality by design: a review. Drug Deliv 2021, 28, 1342-1355, doi:10.1080/10717544.2021.1943056.
- Zhao, W.; Li, J.; Jin, K.; Liu, W.; Qiu, X.; Li, C. Fabrication of functional PLGA-based electrospun scaffolds and their applications in biomedical engineering. Mater Sci Eng C Mater Biol Appl 2016, 59, 1181-1194, doi:10.1016/j.msec.2015.11.026.
- Sonam Dongsar, T.; Tsering Dongsar, T.; Molugulu, N.; Annadurai, S.; Wahab, S.; Gupta, N.; Kesharwani, P. Targeted therapy of breast tumor by PLGA-based nanostructures: The versatile function in doxorubicin delivery. Environ Res 2023, 233, 116455, doi:10.1016/j.envres.2023.116455.
- Butreddy, A.; Gaddam, R.P.; Kommineni, N.; Dudhipala, N.; Voshavar, C. PLGA/PLA-Based Long-Acting Injectable Depot Microspheres in Clinical Use: Production and Characterization Overview for Protein/Peptide Delivery. Int J Mol Sci 2021, 22, doi:10.3390/ijms22168884.
- Zhou, B.; Ma, Y.; Li, L.; Shi, X.; Chen, Z.; Wu, F.; Liu, Y.; Zhang, Z.; Wang, S. Pheophorbide co-encapsulated with Cisplatin in folate-decorated PLGA nanoparticles to treat nasopharyngeal carcinoma: Combination of chemotherapy and photodynamic therapy. Colloids Surf B Biointerfaces 2021, 208, 112100, doi:10.1016/j.colsurfb.2021.112100.
- D'Souza A, A.; Shegokar, R. Polyethylene glycol (PEG): a versatile polymer for pharmaceutical applications. Expert Opin Drug Deliv 2016, 13, 1257-1275, doi:10.1080/17425247.2016.1182485.
- Maadani, A.M.; Salahinejad, E. Performance comparison of PLA- and PLGA-coated porous bioceramic scaffolds: Mechanical, biodegradability, bioactivity, delivery and biocompatibility assessments. J Control Release 2022, 351, 1-7, doi:10.1016/j.jconrel.2022.09.022.
- Abd El-Hack, M.E.; El-Saadony, M.T.; Shafi, M.E.; Zabermawi, N.M.; Arif, M.; Batiha, G.E.; Khafaga, A.F.; Abd El-Hakim, Y.M.; Al-Sagheer, A.A. Antimicrobial and antioxidant properties of chitosan and its derivatives and their applications: A review. Int J Biol Macromol 2020, 164, 2726-2744, doi:10.1016/j.ijbiomac.2020.08.153.
- Sheffey, V.V.; Siew, E.B.; Tanner, E.E.L.; Eniola-Adefeso, O. PLGA's Plight and the Role of Stealth Surface Modification Strategies in Its Use for Intravenous Particulate Drug Delivery. Adv Healthc Mater 2022, 11, e2101536, doi:10.1002/adhm.202101536.

Reviewer 2 Report
Comments and Suggestions for Authors
The authors reviewed the application of PLGA in tumor immunotherapy. The review was written through the search of papers related to PLGA and cancer therapy. The method of obtaining papers was clearly described and the number of papers was adequate for a review. However, significant revisions are required for the review to meet the standards necessary for publication. Below are a few points that need to be modified and improved.
1) The abstract of the review lacks the purpose of the review. The authors just listed out the application of PLGA and there’s no focus point of this review. As a result, readers may struggle to grasp the main idea.
2) The review discussed many different types of PLGA such as the polymer, the nanoparticle form of PLGA, and the microparticle form of PLGA. It will be beneficial for the readers to have a paragraph in the introduction section to briefly introduce differences in size and properties of those PLGA forms.
3) The focus of the review is on the immunotherapy type of cancer treatment, therefore, a paragraph to introduce the mechanism and drugs, as well as limitations of the current immunotherapy is required in the introduction section.
4) The review was written with many redundant paragraphs. For example, line 58 to line 68, line 85 to line 92.
5) The review contains many self-repeating paragraphs, such as line 81 to line 84 is simply repeated in section 4.9. Bibliometrics, line 556 to line 563.
6) The review contains errors such as Section 3.4., line 392, it should be modified for the keywords of search.
7) The data availability statement was just simply from the journal format. Please proofread before sending it to the journal.
8) Figure 5 is inadequately drawn and lacks substantial information that could provide value to the readers.
9) All figure and table captions need to be more elaborate.
Comments on the Quality of English LanguageThe review was written with many redundant paragraphs. For example, line 58 to line 68, line 85 to line 92.
The review contains many self-repeating paragraphs, such as line 81 to line 84 is simply repeated in section 4.9. Bibliometrics, line 556 to line 563.
Author Response
Response to Reviewer 2 Comments
- Summary
Thank you for your letter and the reviewers’ comments concerning our manuscript entitled “Application of PLGA in tumor immunotherapy”. Those comments are valuable and very helpful. We have read through comments carefully and have made corrections. Based on the instructions provided in your letter, we uploaded the file of the revised manuscript.
- Point-by-point response to Comments and Suggestions for Authors
Comments 1.The abstract of the review lacks the purpose of the review. The authors just listed out the application of PLGA and there’s no focus point of this review. As a result, readers may struggle to grasp the main idea.
Response:Thank you for underlining this deficiency. According to the reviewer’s comment, we have provided the purpose of the review in the abstract., which is presented following “We aim to highlight the recent advances and challenges of plga in the field of oncology therapy to stimulate further research and development of innovative PLGA-based approaches, and more effective and personalized cancer therapies.”
Comments 2. The review discussed many different types of PLGA such as the polymer, the nanoparticle form of PLGA, and the microparticle form of PLGA. It will be beneficial for the readers to have a paragraph in the introduction section to briefly introduce differences in size and properties of those PLGA forms.
Response:We deeply appreciate the reviewer’s suggestion. According to the reviewer’s comment, We provided more explanation on the difference between PLGA Nanoparticles PLGA Microparticles PLGA Nanopolymers in the introduction section, line 97 to line 104.
Comments 3. The focus of the review is on the immunotherapy type of cancer treatment, therefore, a paragraph to introduce the mechanism and drugs, as well as limitations of the current immunotherapy is required in the introduction section.
Response:Thank you for your careful review. We added and integrated with the previous paragraph on PLGA tumor immunotherapy a paragraph on the mechanism and drugs, as well as limitations of the current immunotherapy in lines 62 to 77.
Comments 4. The review was written with many redundant paragraphs. For example, line 58 to line 68, line 85 to line 92.
Response:Thank you for underlining this deficiency. This section was revised and modified according to the information showed in the work suggested by the reviewer(lines 57 to 61, 105 to 110). We may have some redundant words when writing the introduction, and we are very sorry for the trouble caused to you.
Comments 5. The review contains many self-repeating paragraphs, such as line 81 to line 84 is simply repeated in section 4.9. Bibliometrics, line 556 to line 563.
Response: We agree with the comment and removed the repetitive sentences of line 81 to line 84. because we planned to put this paragraph of bibliometric analysis in the original article lines 81 to 84, we later deleted this paragraph of bibliometric analysis at the end of the article, and forgot to delete the first part. We are very sorry for the trouble caused to the reading.
Comments 6. The review contains errors such as Section 3.4., line 392, it should be modified for the keywords of search.
Response: We are extremely grateful to reviewer for pointing out this problem. We have… and re-wrote the sentence in the revised manuscript as the following: We have changed the search mode of this part to PLGA (Topic) AND modifi* (Topic) in line 564.
Comments 7. The data availability statement was just simply from the journal format. Please proofread before sending it to the journal.
Response: Our deepest gratitude goes to you for your careful work and thoughtful suggestions that have helped improve this paper substantially. The precedent version of the data availability statement has been replaced, becoming “The data used in our bibliometrics are all derived from results retrieved in web of science in April 2024.”, lines 892 to 893.
Comments 8. Figure 5 is inadequately drawn and lacks substantial information that could provide value to the readers.
Response: Thank you for your precious comments and advice, and we have removed Figure 5. Meanwhile, New figure 3 was added to the Surface modification of PLGA section to give the reader a better understanding of the modification mechanism.
Comments 9. All figure and table captions need to be more elaborate.
Response: We appreciate the reviewer’s positive evaluation of our work and agree with the comments regarding the limitations of our study, and elaborate All figure and table in more detail.
After careful consideration of the reviewer's suggestions, we have thoroughly polished the language of our manuscript. We believe that after this revision, the expression of the paper is clearer, the logic is more rigorous, and the language is more in line with academic standards.
Specifically, we conducted the following:
The entire paper was carefully language proofread to ensure correct grammar.
Some sentence structures were optimized to improve the readability of the article.
Technical terms and representations were standardized to ensure consistency with common terminology in the field.
Thank you for your careful review. We really appreciate your efforts in reviewing our manuscript during this unprecedented and challenging time. We wish good health to you, your family, and community. Your careful review has helped to make our study clearer and more comprehensive.

Reviewer 3 Report
Comments and Suggestions for Authors
This article reviews the “Application of PLGA in tumor immunotherapy”. The manuscript needs improvements as shown below:
1. The explanation of the graphs in Figures 2 and 4 is insufficient. In the graph shown in figure 2, the Y axis is not labeled. Also, the correlation between the yellow graph and the orange bar graph and its meaning are not explained.
2. The word immunocompatible cells does not seem to be commonly used. It can be replaced with the word immunocompatible cell membrane.
3. Line 248, In the sentence “Stimulating human immune regulation,” stimulation and regulation are completely opposite concepts in the immune response. Therefore, the sentence should be changed to something like “regulates the immune response.” Additionally, all similar sentences in the text should be revised.
4. The paragraph containing Table 1 and Figure 3 mainly explains PLGA for immunotherapy. However, the content that Table 1 and Figure 3 are trying to explain do not seem to be significantly related to the paragraph they belong to. The positions of Table 1 and Figure 3 need to be adjusted.
5. The titles of all figure and table legends are too concise. The title should be modified to contain more specific content.
6. All graphs analyzing bibliometrics do not specify the intended meaning they are meant to convey. Beyond simply presenting graphs, opinions derived through analysis must be conveyed to readers.
Comments on the Quality of English LanguageIt needs some modifications.
Author Response
Response to Reviewer 3 Comments
- Summary
Thank you for your letter and the reviewers’ comments concerning our manuscript entitled “Application of PLGA in tumor immunotherapy”. Those comments are valuable and very helpful. We have read through comments carefully and have made corrections. Based on the instructions provided in your letter, we uploaded the file of the revised manuscript.
- Point-by-point response to Comments and Suggestions for Authors
Comments 1. The explanation of the graphs in Figures 2 and 4 is insufficient. In the graph shown in figure 2, the Y axis is not labeled. Also, the correlation between the yellow graph and the orange bar graph and its meaning are not explained.
Response:We are extremely grateful to reviewer for pointing out this problem. We have re-analyzed and displayed the two statistical charts figure 2 and 4, and added the Y axis title and a more detailed explanation of Figures 2 and 4. The yellow line in the previous figure represents the percentage of cumulative frequency, which we feel does not provide information to the reader, so the yellow line is removed.
Comments 2. The word immunocompatible cells does not seem to be commonly used. It can be replaced with the word immunocompatible cell membrane.
Response:Thank you for underlining this deficiency. This section was revised and modified according to the information showed in the work suggested by the reviewer about replacing the phrase immunocompatible cells with immunocompatible cell membrane in line 319.
Comments 3. Line 248, In the sentence “Stimulating human immune regulation,” stimulation and regulation are completely opposite concepts in the immune response. Therefore, the sentence should be changed to something like “regulates the immune response”. Additionally, all similar sentences in the text should be revised.
Response:We deeply appreciate the reviewer’s suggestion. We are sorry that we did not understand the relationship between stimulation and regulation, which caused your reading trouble. We have replaced the phrase “Stimulating human immune regulation” with “regulates the immune response”. All similar sentences in the article were revised in line 123, 140, 168, 349 and 372.
Comments 4. The paragraph containing Table 1 and Figure 3 mainly explains PLGA for immunotherapy. However, the content that Table 1 and Figure 3 are trying to explain do not seem to be significantly related to the paragraph they belong to. The positions of Table 1 and Figure 3 need to be adjusted.
Response:Thank you for your precious comments and advice. Those comments are all valuable and very helpful for revising and improving our paper. and We added a part of some paragraphs about methods of PLGA production in lines 510 to 561, so as to facilitate the context of the article and adjust the position of Table 1 and Figure 3 to this part.
Comments 5. The titles of all figure and table legends are too concise. The title should be modified to contain more specific content.
Response: We appreciate the reviewer’s positive evaluation of our work and agree with the comments regarding the limitations of our study, and elaborate All figure and table in more detail.
Comments 6. All graphs analyzing bibliometrics do not specify the intended meaning they are meant to convey. Beyond simply presenting graphs, opinions derived through analysis must be conveyed to readers.
Response: We deeply appreciate the reviewer’s suggestion. and re-wrote the sentence in the revised manuscript as the following: We explained opinions in the title description and relevant paragraphs of the bibliometric statistical chart.
After careful consideration of the reviewer's suggestions, we have thoroughly polished the language of our manuscript. We believe that after this revision, the expression of the paper is clearer, the logic is more rigorous, and the language is more in line with academic standards.
Specifically, we conducted the following:
The entire paper was carefully language proofread to ensure correct grammar.
Some sentence structures were optimized to improve the readability of the article.
Technical terms and representations were standardized to ensure consistency with common terminology in the field.
Thank you for your careful review. We really appreciate your efforts in reviewing our manuscript during this unprecedented and challenging time. We wish good health to you, your family, and community. Your careful review has helped to make our study clearer and more comprehensive.

Round 2
Reviewer 1 Report
Comments and Suggestions for Authors
Accept.
Reviewer 2 Report
Comments and Suggestions for Authors
The author has revised all the addressed issues.